# Coral tissue depth reconstructed using skeletal microstructural offsets is driven by environmental stress
James Vincent ✉ & Tom Sheldrake ✉

Coral tissue depth reflects organismal health and is influenced by environmental stressors. Reconstructing its past variability on inter- and intra-annual timescales, however, is not yet possible. Here we reconstructed seasonal tissue depth by measuring spatial offsets between growth cycles in corallite porosity and theca geochemistry (Lithium/Magnesium and Barium/Calcium ratios) of a single *Siderastrea siderea* core collected in Barbados. We show spatial offsets and thus tissue depth vary systematically over multiyear timescales, with decreasing values associated with thermal stress that impact extension rate and calcification in subsequent growth cycles. Large environmental disturbances such as the 2021 volcanic eruption of La Soufrière (St. Vincent) also impact tissue depth, in this case likely due to the release of bioactive metals upon ash deposition. This study investigates the robustness of the offset signal within a single colony and with further validation across multiple colonies could help reconstruct regional to global environmental and ecological stressors.

Massive scleractinian corals are symbiotic animals and rely predominantly on their photosynthetic zooxanthellae for energy in the form of fixed carbon for growth[1]. Coral growth involves three fundamental processes 1) linear extension of the aragonitic [CaCO$_3$] skeleton, 2) thickening of pre-existing skeleton within the soft tissues and 3) soft tissue maintenance and production[2–4]. The physiology of the coral holobiont and consequently growth is sensitive to changes in oceanic parameters such as sea surface temperature (SST), nutrient availability, and insolation[5–7]. The sensitivity of coral growth to their surrounding environment has been widely exploited, making both modern and fossil corals valuable tools for reconstructing past environments at local, regional, global and geological (time)scales. Radiographs taken perpendicular to the major growth axis of the skeleton are used to reconstruct seasonal variations in the first two processes by assessing annual growth bands[8–10]. These bands are characterised by intervals of high and low skeletal densities which generally reflect variability in the rates of coral thickening and extension[4,11]. Generally, by measuring the distance between high density bands one can reconstruct the annual linear extension rate[12,13]. For massive, porous corals in which soft tissue penetrates into the skeleton, higher density bands are associated with slower extension rates in relatively cooler SSTs and lower calcification rates[5,12]. These bands form the basis for chronologically backdating coral growth and are often used alongside geochemical proxies such as trace elements that substitute for Ca$^{2+}$ in relation to certain environmental parameters (i.e., Li/Mg and Ba/Ca ratios for SST and terrestrial input, respectively) to infer information about past environmental conditions[14,15]. Hence, low density bands are associated with

faster extension rates, which are typically associated with warm SSTs when calcification rates are higher.

Environmental changes can induce stress on coral reef ecosystems, which may induce physiological changes that disrupt coral homeostasis. Large-scale negative stressors such as ocean acidification and increasing SST associated with green-house gas emissions[16] decrease coral calcification rates[17] and disturb coral symbiosis with zooxanthellae[18]. The latter often leading to coral bleaching, whereby the corals expel their symbionts reducing their primary source of energy[18]. Local or regional stressors such as reefs situated proximal to rivers[19,20] and around volcanic islands (i.e., episodic explosive eruptions) can harm corals by increasing sedimentation, turbidity, and nutrient stress[21–24]. Sedimentation causes particulate matter to settle on coral surfaces, suffocating the coral polyps and hindering heterotrophic feeding[22]. Sedimentation also increases turbidity, which reduces light for symbiont photosynthesis reducing energy acquisition. Corals also expend energy producing mucus to shed sediment, diverting energy away from growth and repair[25,26]. Under negative stressful conditions whereby photosynthetic and heterotrophic feeding are inhibited, corals rely on lipid (wax esters and triacylglycerols) energy stocked in soft tissues to maintain essential physiological processes and coral homeostasis[27–32]. The depletion of lipid reserves consequently reduces the depth of soft tissues which thus serves as a proxy for the health status and resilience of the coral to environmental stress[30,33–35]. Conversely, sedimentation stressors may induce positive responses. For example, it is reported that corals living in coastal environments with high turbidity and sedimentation rates had deeper

Department of Earth Sciences, University of Geneva, Genève, Switzerland. ✉e-mail: james.vincent@unige.ch; thomas.sheldrake@unige.ch

tissues and somatic energy reserves compared to low-turbidity offshore reefs[25]. Such conditions reduce mortality under high temperature and light conditions by alleviating light pressure and providing an alternative food source[36]. Terrestrial material such as volcanic ash or desert dust releases bioactive metals (e.g., Fe and Mn) that enhance coral photophysiology and biomineralisation[37].

Given tissue depth (TD) is a good indication of the health of a coral, such a measurement is insightful for understanding the impacts of environmental change. TD is typically measured in situ by measuring the depth in which tissues penetrate the skeleton[13,38,39], and understanding multi-year seasonal changes in coral health requires regular TD measurements throughout the year. Repeatedly sampling the same colony, however, is invasive, compromises coral health, is time-consuming and requires proximity to the site. Alternatively, some massive forms of corals form dissepiments, which are thin, horizontal sheets of aragonitic skeleton that fuse between vertical structures perpendicular to the major growth axis[4,13,40]. Dissepiments act as step for the polyp and soft tissues to ascend, with new steps forming at approximately monthly intervals. Assuming that as the coral extends soft tissues are uplifted at the same time as dissepiment formation, this enables past TD to be reconstructed by measuring dissepiment spacing[33]. An important limitation to this method, however, is that not all corals form dissepiments, and for those species which do, identifying them can be difficult[4]. Additionally, it is complicated to deconvolve dissepiment spacing from polyp accent rates/extension rates and TD which are interlinked but not correlated[13]. Therefore, it is only possible to measure the TD at the time of collection, meaning any record of TD (i.e., coral health) throughout growth is difficult to calculate.

Soft tissues penetrate the first few mm of skeleton at the growth surface and, depending on the extension rate, can overlap with the previous growth band, causing inconsistencies between interpretations[41]. This overlap of soft tissues and mass accumulation between the current and previous growth band results in bio-smoothing, whereby skeletal thickening overwrites and buries the original seasonal signal[38,39,42]. For example, the mean tissue depth (TD) of common Caribbean coral *Siderastrea siderea* was measured at $6.35 \pm 0.14$ mm, with an annual extension rate (i.e., band width) of $3.54 \pm 0.14$ mm/yr ($n = 35$)[38,39,42]. Tissues therefore overlapped with approximately half the previous band potentially bio-smoothing six months of growth records. To add further complication, both extension rates and TD covary intra- and inter-annually in response to environmental stress[5,7,13,25,36,43,44]. TD is therefore an important parameter to constrain when using coral proxies to reconstruct past oceanic parameters.

To overcome the limitations with reconstructing TD, we utilise a previous result for the Caribbean coral *Siderastrea siderea*, which showed that growth bands originate through the thickening of skeletal structures within the corallite (i.e., septa and columella) to a level below the base of the corallite cup[41] (Fig. 1). Since the upper surface of individual corallites of *S. siderea* have a concave structure, the distance between the fully thickening corallite skeleton (yellow star in Fig. 1) and upper thecal surface of the sample (green star in Fig. 1) represents the interval of soft tissue in which calcification is occurring. Hence, by tracking seasonality in corallite porosity and comparing to seasonality in theca geochemistry, it should be possible to track tissue depth.

In this study, we expand upon the results described in the previous paragraph to reconstruct past TD and coral health in response to thermal and volcanic stress. The sample (HP1) was collected during the warm-/wet-season in July 2022 at 15 m water depth from the north-west coast of Barbados (see Supplementary Fig. S.1), following the deposition of 2–5 mm of volcanic ash in April 2021 from La Soufrière volcano on St. Vincent (180 km east of Barbados)[45]. The growth banding pattern of HP1 was revealed by segmenting pixels from two-dimensional y-axis slices reconstructed from μCT scans, to produce a depth profile of porosity within the corallite skeletal structures. Geochemical proxies (Li/Mg and Ba/Ca) were measured along the theca of four individual corallites using LA-ICP-MS and were used to reconstruct seasonality. Li/Mg is reported to track seasonal variations in SST[15,46,47] whilst Ba/Ca ratios have been used to assess terrestrial

runoff[48], upwelling[49], riverine/fluvial discharge[50], and precipitation[51]. This study uses Ba/Ca to represent the seasonal rainy season in the Caribbean[14,52] which coincides with elevated SST in summer (see methods and Supplementary Figs. S.2–S.5). We spatially synchronised the growth and geochemical cycles and measured the peak and trough positions (see Methods). We show that the peak and trough positions of the corallite porosity and theca geochemical cycles are systematically offset, and that this offset is inversely correlated with bleaching HotSpot values, a direct indicator of coral stress that can influence future calcification. We also show that the offset distance is also driven by more local and regional stressors, using the exposure to volcanic ash from the 2021 La Soufrière (St. Vincent) eruption as an example. This case study introduces a novel approach to reconstruct tissue depth inter-/intra-annually and to assess coral health (i.e., stress tolerance) in response to external stressors.

## Results and discussion
### Offsets between coral element ratios and porosity reflects tissue depth
The raw geochemical data (grey circles) and smoothed Li/Mg (solid red line) and Ba/Ca (solid blue line) ratios are shown in Fig. 2. Both the smoothed Li/Mg and Ba/Ca profiles show fluctuations resulting in nine clear cycles. LOESS-smoothed median Li/Mg values vary between 1.07 and 1.24 mmol/mol with a mean of 1.16 mmol/mol, whilst Ba/Ca ranges between 5.33 and 22.27 μmol/mol with a mean of 9.83 μmol/mol. The dashed lines in Fig. 2 illustrate the confidence band, representing the 5–95th percentile range of the bootstrapped LOESS fits. For both Li/Mg and Ba/Ca, the confidence bands show that whilst the amplitude of cycles (i.e., the y-axis) may vary because of the analytical error of the measurements, the spatial position (x-axis) of the peak and troughs remain largely unchanged.

The reconstructed corallite porosity (black line) is plotted against the LOESS-smoothed theca Li/Mg and Ba/Ca (red and blue lines, respectively) in Fig. 3. Corallites are fully thickened at 3.8 mm (yellow star—Figs. 1, 3) beneath the uppermost surface of the sample (black vertical line – Fig.3). Nine cycles corresponding to nine annual growth bands are observed in the corallite porosity reconstruction. The first fully formed growth band is a low-porosity band (i.e., trough) at 5.9 mm (orange star, Figs. 1, 3) below the uppermost growth surface. Given the sample was collected in July 2022, transitioning from cold-/dry- (C/D) season to warm-/wet- (W/W) season, we interpret the closest porosity trough to the growth surface (orange star— Fig. 3) to represent the C/D-season with low SST's when investment of energy into linear skeletal extension is expected to be lower[4,5,12].

At first glance of Fig. 3, it may appear that low values in Li/Mg (i.e., warmer SST) are associated with lower values of porosity, but this is not the case due to the geometrical complexities of the upper surface of the coral sample (Fig. 1). If such geometrical complexities are not accounted for, it is possible for seasonal variability to be misinterpreted. With this perspective, we can see that the closest porosity trough to the growth surface is offset, but corresponds well to the theca geochemistry, in which the Li/Mg peak and the Ba/Ca trough closest to the growth surface corresponds to the low SST's[46] and low terrestrial run-off[14,52] characteristic of the C/D-season. On this basis, high porosity bands correspond to the W/W-seasons characterised by fast skeletal extension rates (ranging between 3.56 to 5.17 mm/season), whilst low porosity bands correspond to C/D-seasons with slower extension rates varying between 1.25 and 2.17 mm/season (Fig. 4). Chronologically back-dating these cycles reconstructs the sample's age from W/W-season 2022 to W/W-season 2013. The annual extension rates between the porosity peaks varies between 5.19 and 6.44 mm/yr, with a mean extension rate of 5.89 mm/yr).

Spatially synchronising the Li/Mg and Ba/Ca ratios of the theca to the corallite porosity reconstruction reveals a systematically changing negative offset indicated by the pink coloured arrows in Fig. 3. The porosity troughs and corresponding Li/Mg peaks and Ba/Ca troughs are hereafter termed C/D-season offsets. The porosity peaks show a similar negative offset with Li/Mg and Ba/Ca cycles which is hereafter termed W/W-season offsets. Both C/D- and W/W- seasonal offsets show

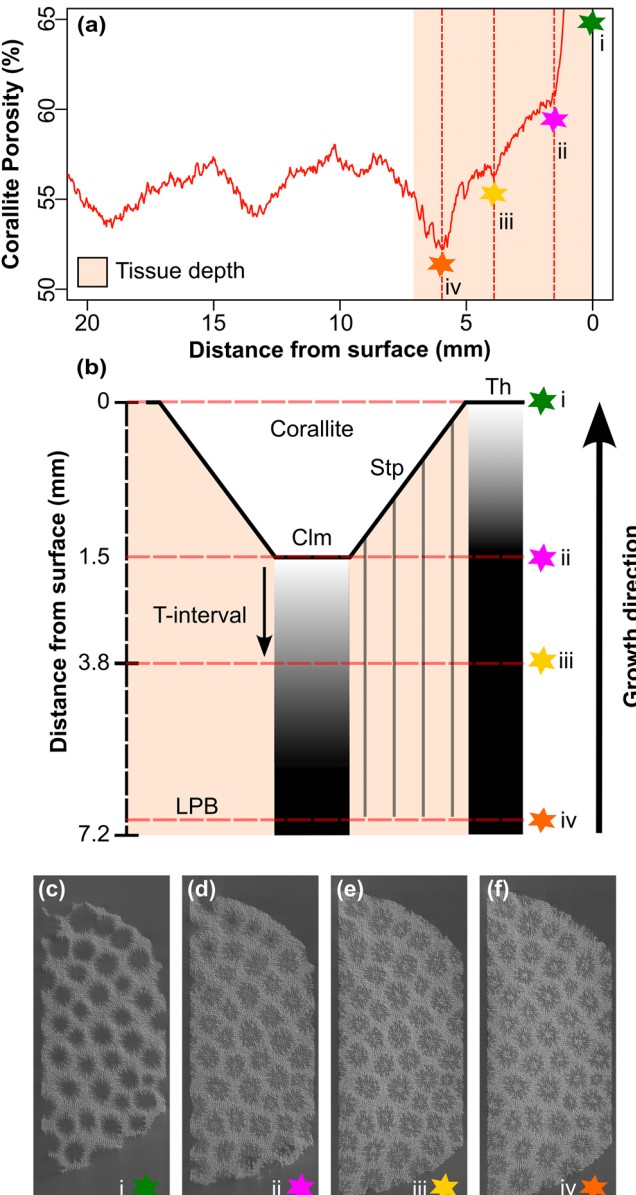

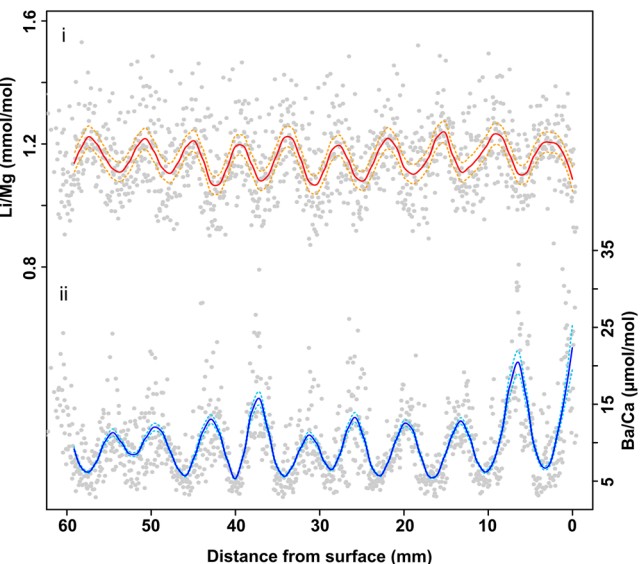

**Fig. 2 | Raw and smoothed geochemical profiles of Li/Mg and Ba/Ca ratios of HP1.** The raw datapoints from all analytical sessions (T1-T4) are plotted as grey dots in the Li/Mg and Ba/Ca profiles. The solid red (i) and blue (ii) lines correspond to the LOESS-smoothed median value of the bootstrapped ($n = 50,000$) dataset for Li/Mg and Ba/Ca respectively, whereby individual datapoints were sampled according to the analytical error calculated using the JCp-1-NP reference material. The dashed orange and light-blue lines, correspond to the 5th and 95th percentile limits of the bootstrapped dataset. Note that for Ba/Ca, the 5th and 95th percentile limits plot closely with the median value and that in both profiles, the analytical error does not influence the x-axis position of the peak and troughs of the cycles. The analytical precision does however influence the amplitude of the cycles which is more clearly observed in the Li/Mg profile. See Supplementary Figs. S.7, S.8 for more details.

The offsets between theca Li/Mg and corallite porosity and Ba/Ca and corallite porosity correspond with microstructural evidence of skeletal thickening down to 3.8 mm[41] (Fig. 1) within soft tissues. This measured offset is less than the depth of tissue staining at the surface of the sample (see Supplementary Fig. S.1), which implies the staining depth includes organic material below the depth of actively calcifying tissue, such as tissue undergoing necrosis or other endolithic species. Given the positive relationship between SST and calcification rate[4,7], it is more likely that the distance between cycles of Li/Mg and corallite porosity represents the true TD, although the bimodality of the porosity peaks also reflects a similar feature in the annual rainfall in Barbados (see Methods). Comparing the distance between the first porosity trough (orange star–Figs. 1, 3) and the first Li/Mg peak (Fig. 3), 3.7 mm, to initial thickening offset between the growth surface and fully thickened corallite - 3.8 mm (black arrow and yellow star in Figs. 1, 3), provides further evidence to suggest the Li/Mg offset reflects the depth within soft tissues where the skeleton is thickened. Ba/Ca follows similar offset trends as Li/Mg (Figs. 3, 4a) which indicates that changes in the offset are not driven by the timing of the process driving the geochemical parameter, given that Ba/Ca and Li/Mg are spatially offset from each other and thus controlled by different environmental processes.

To ensure the robustness of the offset between theca geochemistry and corallite porosity, we measured four different theca transects for HP1 (see Supplementary Fig. S.6), given that the geochemistry of *S. siderea* is known to vary horizontally[56]. The horizontal geochemical variation is observed when comparing the geochemistry of individual transects (see Supplementary Figs. S.7, S.8). Whilst the individual transects show the same seasonal signal, they are slightly offset from each other (see Supplementary Fig. S.7). The offsets between the geochemical profiles results in poor correlation between the individual theca, especially at low spatial resolutions (Supplementary Fig. S.8). Even at high spatial resolution these geochemical offsets between transects exist, suggesting that they might result from subtle

**Fig. 1 | Corallite skeletal thickening interval. a** The upper growth surface porosity profile of HP1 illustrating four stages of skeletal formation (adapted from Vincent & Sheldrake, 2025[41]). **b** Schematic of skeletal formation stages. **c**–**f** Micro-CT y-axis slices of skeletal formation. The four stages of skeletal formation are shown schematically in (**b**) and by CT slices in (**c**–**f**). The reconstruction start (0 mm) was adjusted and defined between the first reconstructed skeleton and the fully reconstructed corallite (slice #2193). Green (i), pink (ii), yellow (iii), and orange (iv) stars mark the growth surface, base of the corallite cups (septa and columella), the uppermost level where the skeleton is fully thickened, and the nearest low-porosity band (cold/dry-season), respectively. Orange shading indicates tissue staining depth measured upon sample collection (7.17 mm). The schematic (**b**) shows corallite structures; theca (th), septa (Stp), and columella (Clm), and the black arrow (T-interval) highlights the depth at which corallite skeleton is thickened (i.e., skeletal thickening interval).

cyclical variations over approximately 3 to 4-year cycles (Fig. 4a). The peaks and troughs in Ba/Ca cycles occur after Li/Mg cycles, which have larger offsets (3–3.7 mm) compared to the Ba/Ca (2.2–3.2 mm). The difference between Li/Mg and Ba/Ca offsets may correspond to differences in the timing of rainfall and SST in Barbados[53] (see Supplementary Figs. S.3–S.5) but is likely influenced by surface hydrological and nutrient supply processes[54,55].

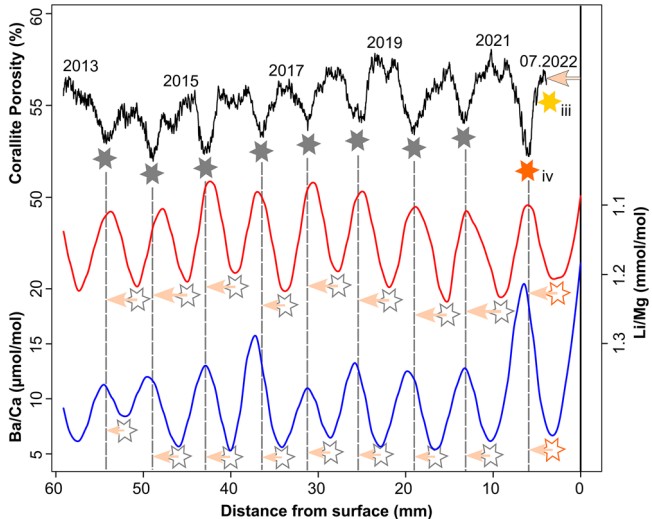

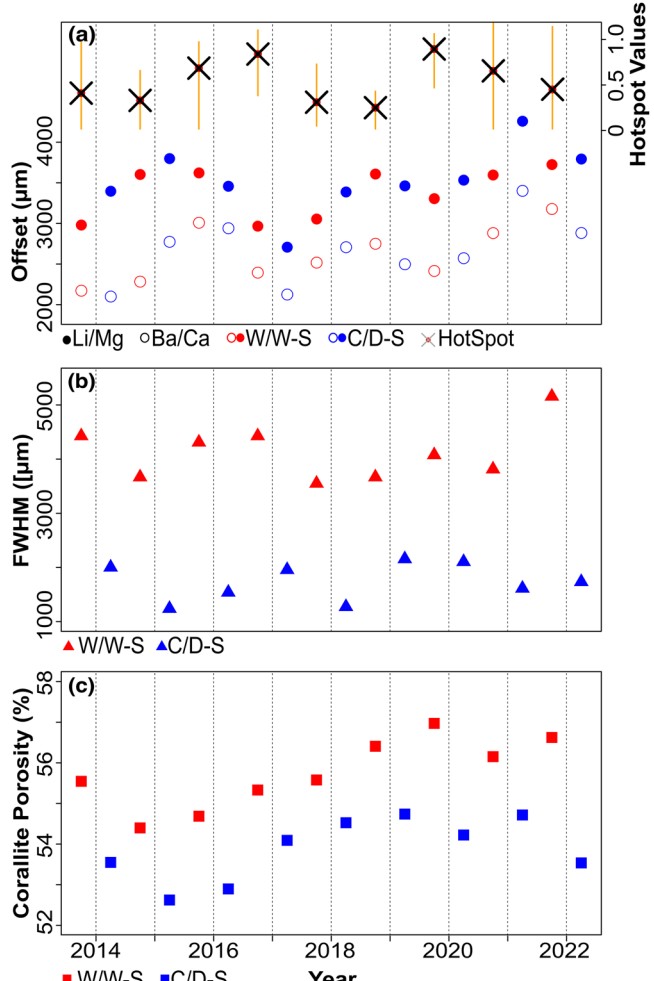

**Fig. 3 | Spatially synchronised porosity and LOESS-smoothed Li/Mg and Ba/Ca profiles.** Corallite porosity, theca Li/Mg, and theca Ba/Ca ratios of HP1 correspond to the black, red, and blue lines, respectively. The porosity and element profiles show 9 annual cycles corresponding to 9 years of growth from warm/wet-season in 2022 (i.e., time of collection) to the warm/wet-season in 2013. The black vertical solid line corresponds to the uppermost theca surface of the sample that is time synchronous to the fully thickened corallites (yellow star [iii] in Fig.1). The first low porosity band (orange star [iv] marks the preceding cold/dry-season growth band. The grey stars indicate the depths of the porosity troughs corresponding to cold/dry-season, with the unfilled stars corresponding to the same cold/dry-seasons in the Li/Mg and Ba/Ca profiles. The dashed vertical grey lines extend these positions to the chemical profiles to show the offset (i.e., tissue depth) with the troughs in the chemical profiles (note that Li/Mg has been inversed for clearer comparability). The orange arrow outlined in black represents the interpreted tissue depth of active skeletal thickening at the growth surface (3.8 mm). The size of the orange arrows indicates the direction and distance of offset between the porosity (i.e., grey dashed line) and the chemical profiles. The arrow size is different for Li/Mg and Ba/Ca as these two geochemical signals are offset with each other as they result from different seasonal parameters.

**Fig. 4 | Multi-year seasonal record of tissue depth, extension rates and porosities. a** shows a timeseries of the spatial offsets between the warm/wet- and cold/dry-seasons (red and blue points, respectively) in Li/Mg (solid points) and Ba/Ca (hollow points) ratios and the corresponding corallite porosity cycles. The vertical dashed lines represent the boundary between each calendar year. The orange boxes in (**a**) represents the range of the 90th IQR of seasonal HotSpot values from the NOAA Coral Reef Watch. The solid red circles with the black crosses correspond to the median seasonal HotSpot value which is used to depict inter-annual trends. **b** shows a timeseries of warm/wet- and cold/dry-season FWHMs (i.e., seasonal extension rates, represented by the red and blue triangles, respectively) measured on the intra-corallite porosity cycles between warm/wet-season 2013 and cold/dry-season 2022 (illustrated by the horizontal grey dashed lines). **c** shows a timeseries of the seasonal porosities corresponding to the same porosity cycles used to measure the FWHM in (**b**). Warm/wet- and cold/dry-seasons are represented by the red and blue squares respectively.

variations in the growth surface such that it is not perfectly planar. Consequently, we combined data from multiple transects until the sample average geochemistry can be calculated. By progressively combining transects we converge towards the true sample average, with fewer transects needed if the sampling resolution per transect is higher (see Supplementary Fig. S.8). We are therefore confident that the LOESS-smoothed median of all combined 4 transects (1200 datapoints) represents the true average signal of the entire sample. These results highlight the importance of multi-theca (total sample) analyses when comparing with the sample porosity, as well as the advantage of LA-ICP-MS that allows for high resolution sampling of individual theca structures. In the future, for an individual colony this could be overcome by matching individual corallite porosity and geochemistry but would still require multiple corallite measurements to track the overall stress state of the coral organism.

## Tissue depth is responsive to environmental stress

Coral TD is reported to vary in response to environmental parameters and stress[13,25,33,34,57], which is the supposed driving force of the change in the theca geochemistry – corallite porosity offset (Fig. 4a). In this study, thermal stress accumulation (as indicated by the HotSpot values in Figs. 4a, 5; see methods) was elevated during 2015, 2016, 2019 and 2020. The general relationship between TD and thermal stress (i.e., offsets and HotSpot values) is shown in Figure 5, with years experiencing high heat stress located to the top left and associated with decreasing spatial offsets. A decrease in the spatial offset is interpreted as reduced TD, thus decreasing energy reserves in soft tissues[27–30,35,36]. During such periods of thermal stress, energy acquisition via photosynthesis is compromised[27–30]. Without energy supply via symbionts, or sufficient SPM for heterotrophic feeding, coral lipids stored in existing

tissue will be depleted to supply the coral with energy for respiration and skeletal extension[2,3,36]. The depletion of lipids decreases TD and subsequently reduces the vertical distance between the theca Li/Mg and Ba/Ca ratios and the depth at which skeleton is thickened which is responsible for the observed offset. Conversely, years with less accumulated heat stress (e.g., 2014 and 2018) show increasing spatial offsets implying soft tissues were thickening, with energy stores replenished during a recovery phase as conditions improved for symbiotic photosynthesis. Such increases in TD imply larger energy reserves that reduce the risk of starvation during coral stress[26,28,30,31] making TD an important stress response indicator[30,33,58]. Whilst lipid levels, and consequently TD, can vary seasonally (with light intensity and SST[34–36,44]) due to energy production from photosynthesis, the lack of seasonal signal in the offset (Fig. 4a) suggests this is not the case for *Siderastrea siderea*.

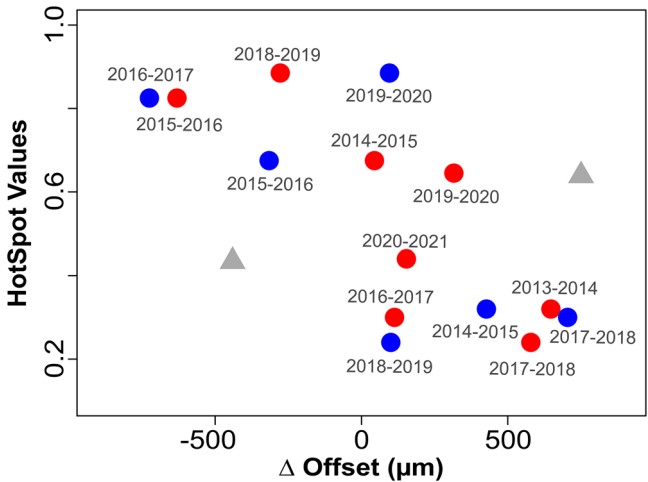

**Fig. 5 | Scatter plot showing the relationship between the Δ offset and HotSpot values.** Red and blue circles correspond to the Δ offset values between consecutive years for warm/wet-seasons and dry/cold-seasons, respectively. For example, for 2016 the red circle represents the change in offset distance between the warm/wet season of 2015 and warm/wet season of 2016; and the blue circle represents the change in offset distance between the cold/dry season of 2016 and cold/dry season of 2017. The Δ offset is plotted against the HotSpot values corresponding to the years labelled in grey. The grey triangles represent years in which the Δ offset included anomalous measurements from 2021 cold/dry season which are influenced by the volcanic eruption. Years with elevated accumulated heat stress are 2015, 2016, 2019, and 2020, which are situated towards the top left of the scatter plot. Horizontal scatter is the points likely reflects the limitations of measuring the offset at a seasonal resolution, as well as the influence of more local or regional stressors, which is observed when considering the points related to the volcanic eruption in 2021.

In April 2021 the north-west coast of Barbados experienced ashfall associated with the eruption of La Soufrière on the island of St. Vincent, almost 200 km due west[59]. Over a period of thirteen days a sequence of forty explosive events ejected volcanic ash more than 15 km into the stratosphere and deposited 2–5 mm of volcanic ash on and around Barbados[60]. This volcanic event disturbed many environmental parameters important for coral growth (e.g., increased turbidity, sedimentation, insolation, and SST[7,8,61]), resulting in baseline deviations in TD (Fig. 4a), skeletal extension (Fig. 4b) and possibly skeletal porosity (Fig. 4c). In the C/D-season of 2021, concurrent with the La Soufrière eruption, an increase in TD is observed (Fig. 4a), clearly falling out of line with possible longer-term trends driven by thermal heat stress. The La Soufrière eruption is an important example of how local or regional stressors can influence TD, and thus may explain the horizontal scatter in Figure 5. This increase in TD (i.e., increased energy acquisition and energy storage) can be explained by the large-scale release of ocean-limited bioactive metals (e.g., Mn, Fe) from volcanic ash[62]. Volcanic ash surfaces contain soluble salts comprising of essential metals for photosynthesis[63], which are leached upon ash deposition in seawater and known to influence coral photophysiology under aquarium settings[37]. Assuming that ash exposure had similar effects on massive corals in natural-reef environments, the metals released from the ash enhance photosynthetic efficiency leading to an increase in energy storage, as indicated by the increased TD in our study. Alternatively, volcanic ash can also be ingested heterotrophically as SPM, possibly providing an alternative nutrient and energy when photosynthesis is compromised due to the elevated thermal stress[2,25,26], as experienced in high turbidity, inshore reefs. Suspended particulate matter is reported to alleviate stress associated with light pressure and elevated SSTs by providing an alternative source of nutrition[34]. The impact of this event appears short-lived, with TD decreasing in-line with existing trends in the W/W-season of 2021. This short-lived response is likely due to dominant stress being thermally driven stress (Fig. 5) and the distal location of Barbados to St. Vincent limiting the timescales of ash remobilisation and consequent supply of bioactive metals.

## Tissue depth and its impacts on seasonal calcification

Thicker soft tissues and deeper TDs imply greater energy reserves, meaning that energy gained through photosynthesis can be allocated to calcification. For example, changes in the W/W-season offset between the theca Li/Mg and corallite porosity are mirrored by skeletal extension in the W/W of the following year (Fig. 6). This time-dependent behaviour can be statistically separated into two regimes (see Methods): (i) deeper TD followed by faster extension rates (i.e., 2014–15, 2015–16, 2018–19, 2019-20); and (ii) shallower TD followed by slower extension rates (i.e., 2013–14, 2016–17, 2017–18). The one exception is 2020–21, where extension rates exceed preceding TD coincidental to the volcanic eruption (as discussed in the previous section). Years in which faster extension are observed follow years of low or no thermal stress, during which tissues have thickened (Figs. 4a, 6). As TD is driven by negative coral stress (Fig. 4a[28–30,33,34,43,57]), which is typically dictated by W/W-season environmental conditions, this relationship underlines the importance of soft tissues in maintaining coral extension rates inter-annually. The correlation between TD and future extension providing further support for the role of W/W-season physiological state driving inter-annual energy budgets.

The observed year-lag relationship between TD and extension rate indicates that the coral's physiological state does not have an immediate effect on concurrent extension. This lagged relationship could result from quantifying extension at a seasonal scale, whilst heat stress accumulates towards the mid to latter stages of the W/W-season but nevertheless suggests that tissue response to stress is more dynamic than calcification response[13]. There is also a lack of correlation between W/W-season TD and subsequent C/D-season extension, implying that energy stored in soft tissues during the W/W-season is not used for C/D-season extension. Instead, C/D-season extension is correlated to the previous W/W-season porosity (Fig. 7a) suggesting that calcification in the C/D-season is driven by the same process(es) controlling corallite porosity in the W/W-season. Coral porosity exhibits seasonality with higher porosity in the W/W-seasons (Fig. 4c) but follows an underlying larger-magnitude baseline trend that is disconnected from TD and intra-annual variability (Fig. 4a, c). What controls this long-term trend is unclear and requires a more extensive record, but it appears that W/W-season extension is correlated to W/W-season porosity, with parallel linear trends observed for the periods 2013–16 and 2017–20 (Fig. 7b). Although this relationship between extension and porosity has also been reported in literature[61,64–66], the reason for it remains unclear. It could be due to extension limiting the duration of an actively calcifying surface residing within the thickening interval of the soft tissues[40], which may be shorter in times of stress when extension rates are elevated[7,8]. Alternatively, it is possible this relationship is not purely causal and instead implies that C/D-season extension is driven by the longer-term baseline signal observed in porosity, in addition to TD.

## Conclusions

We spatially synchronised Li/Mg and Ba/Ca ratios of the theca to corallite porosity cycles within a single *Siderastrea siderea* sample. The FWHM was measured on Gaussian fitted porosity and chemical cycles to identify seasonal peak and trough positions and to assess variations in offsets between corallite growth cycles and Li/Mg and Ba/Ca theca ratios. Our results show a systematic spatial offset between theca geochemical and corallite porosity cycles that results from the complex surface geometry of corals. Understanding this geometrical complexity is important to ensure seasonal variations in porosity/density are not misinterpreted. The spatial offsets we measure vary through the sample and are interpreted as changes in coral tissue depth, which is in turn related to environmental stress. Negative thermal stress during periods of high SST in the warm/wet-seasons dictates whether the coral consumes energy stores in soft tissues and consequently reduces the tissue depth or acquires and accumulates energy increasing tissue depth. The negative relationship between increased thermal stress and reduced offset exhibits some scatter, likely due to other local and regional environmental stressors. One such clear example is associated with volcanic ash deposited from the 2021 explosive eruption of La Soufrière, St. Vincent,

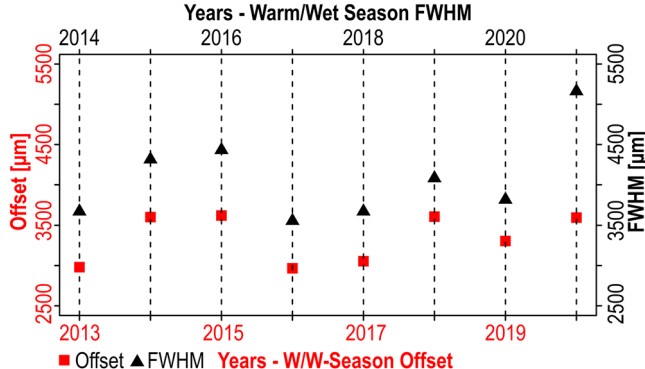

**Fig. 6 | Single year lag in seasonal extension rates with tissue depth.** The graph displays the positive relationship between trends in warm/wet-season tissue depth (i.e., the spatial offsets between the troughs in Li/Mg ratios and the peaks in corallite porosity – red squares) and trends in the warm/wet-season extension rates in the following year (i.e., FWHMs – black triangles). The vertical dashed lines illustrate the year's corresponding to the offset (red) and FWHM (black) noted on the bottom and top x-axis respectively. Note that 2021 has the fastest extension rate in the 9-year record.

which increased tissue depth and skeletal extension rates. We interpret this signal as a positive stress response to bioactive metals released from ash leaching. We reveal a link between tissue depth and future calcification at multi-annual timescales, showing that thermal stress dictates the physiological state of the coral (i.e., consumption or accumulation of energy), with increased skeletal extension associated with tissue thickening.

The findings presented in this case study show the utility of exploiting microstructural offsets to understand variations in tissue depth. Although the robustness of the approach has been investigated at an intra colony scale, we propose that further validation using a multi-colony approach is required to establish the microstructural offset as a tool to quantitatively reconstruct soft tissue depth and coral stressors. Additionally, future research should be taken on more complex corallite structures (W-shaped) and/or more complex growth histories than the single colony presented here. Nevertheless, to the best of our knowledge, this case study provides a new and novel approach to estimating paleo coral health, and the ability to reconstruct past tissue depth and offers an opportunity to examine how marine ecological stress in tropical coral reefs is modulated by both natural and anthropogenic processes.

## Methods
### Geological context and climate
Barbados is an Eastern Caribbean Island located in the North Atlantic and is part of a chain of islands known as the Lesser Antilles. The tropical island is the only exposed part of an accretionary prism complex known as the Barbados Ridge and is geologically independent from the neighbouring volcanic arc to the west. The island is composed of 85 % Pleistocene reef limestones terraces built upon 15% Tertiary pelagic and hemipelagic rocks that outcrops in the Northeast of the island (Scotland District)[67]. The tertiary rocks were uplifted as the island was raised, forming a series of reef terraces. Well-developed, silicate-rich soils are found on the Pleistocene limestone and are made up from predominantly aeolian African dust carried by the trade winds from Africa, volcanic ash from the volcanic island of St. Vincent, and tertiary sandstones and mudstones[68,69]. The most recent explosive eruption since this publication was in April 2021[59] which deposited 2–5 mm of volcanic ash on the island[45]. The initial target of this study was to investigate the volcanic ash impacts on Barbadian reefs. The 97 km of coastline is surrounded by fringing, patch and bank coral reefs with the islands best developed reefs typically on the northwest coast/leeward side[70]. Reefs along the west coast, however, are affected by anthropogenic stressors such as urbanisation/sedimentation, tourism, agriculture, and industries (cement manufacturing, rum distilleries[71,72]).

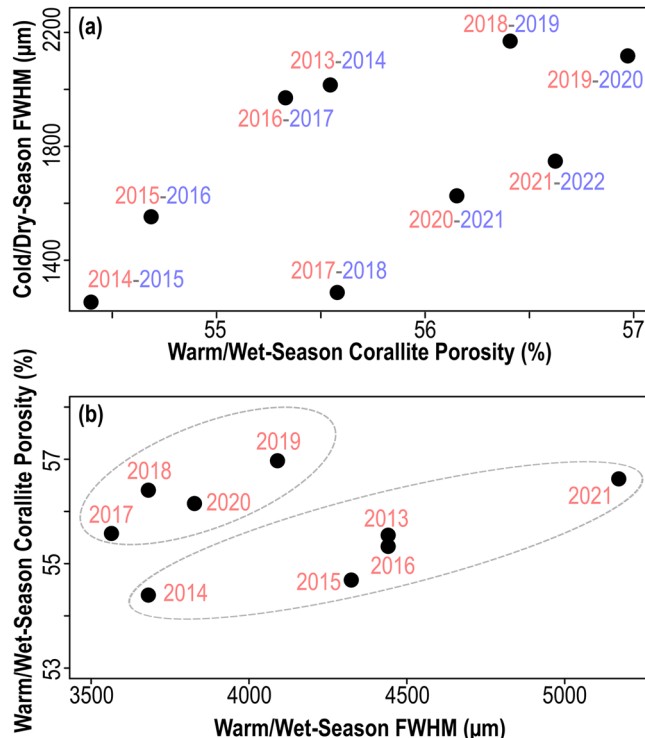

**Fig. 7 | Seasonal extension rates and porosities. a** shows the positive relationship between warm/wet-season porosities and subsequent cold/dry-season extension rates. **b** shows the relationship between warm/wet-season FWHM (i.e., extension rates) and corresponding warm/wet-season porosities. In (**b**), a broad relationship is observed with increases in warm-season FWHM associated with increases in warm-season porosity, especially when separating the two periods 2013–2016 and 2017–2020 (indicated by the dashed grey lines).

Barbados experiences a climate with relatively consistent SSTs throughout the year. Lower SSTs of approximately 25 °C generally occur between February and March whilst warmer SSTs of approximately 29 °C occur between September and November. Consequently, the island experiences warm- and cold- seasons which coincide with the wet- and dry-months. Consequently, between June to November the climate is warm and wet and between December to May (relatively) colder and drier (see Supplementary Figs. S.2–S.5). We therefore term these periods warm/wet (W/W) -seasons and cold/dry (C/D) -seasons. A typical C/D-season is characterised by less rainfall and reduced humidity than in the W/W-season which in turn experiences higher rainfall and greater humidity, with also increased frequency of tropical cyclones. Annual rainfall in the Caribbean shows a distinct bimodal behaviour, with the first mode termed "the early rainfall season" between April to July and the second mode termed "late rainfall season" from August to November[73,74]. Between the two modes is referred to as the "midsummer drought" which is believed to be correlated to vertical wind shear and atmospheric particles such as Sahara dust[75,76]. The islands climate and weather are influenced by El Niño-Southern Oscillation (ENSO) events. During El Niño, Barbados tends to experience drier-than-normal conditions that can lead to decreased rainfall, decreased tropical cyclone activity, and warmer SSTs[77]. Conversely, during La Niña events, Barbados typically experiences wetter-than-normal conditions, cooler SSTs, and increased tropical storms[53] and references there-in). La Niña events were observed in 2016–2018 and 2020–2021 whilst El Niño events were recorded in 2014–2016 and 2018–2019.

The seawater surrounding Barbados is influenced anticyclonic eddies known as the north Brazilian current (NBC) rings that transport warm, nutrient rich, low saline waters from the Amazon and Orinoco rivers[78,79] towards the Caribbean. Rings develop from retroflection of the NBC at irregular intervals, typically between 4 to 5 times a year, with a peak

frequency between December and January[80,81]. The rings in boreal winter are larger, faster rotating, more energetic and shorter lived relative to boreal summer and early autumn[82]. Migrating north-westward along the South American coast, the rings interact with the Lesser Antilles volcanic arc and Barbados[80]. The duration of their interaction with Barbados coastal waters varies depending on ring-type and the angle at which these rings intercept the island but can last up to 150 days[79,83]. These intrusions are associated with green water masses several times a year around Barbados and variable water current speeds and directions, which may reduce coral susceptibility to heat stress by enhancing water flow, lowering irradiance, and increasing plankton availability for heterotrophic feeding[79]. The transportation of warm, low-salinity, nutrient-rich, turbid water bodies, however, likely influences the skeleton Ba/Ca ratio, which is interpreted as a tracer for nutrients in upwelling regions[84]. The precise cause of the Ba/Ca signal however is not the focus of this study.

## Collection and preparation
A cylindrical core 45 mm in diameter and 66 mm in length was extracted from a *Siderastrea siderea* colony growing at 15 m water depth from a forereef setting in July 2022 during the W/W-season. The sample was located away from populated and touristic areas and industry off the coast of Harrison's Point on the northwest coast of Barbados (N 13°18.3288 W 059°39.5403) (see Supplementary Fig. S.1). The sample is named HP1. In the work of Vincent & Sheldrake (2025)[41], this sample was first investigated alongside a second sample collected at a shallower depth of 5 m (W1) to develop a method to quantify skeletal porosity using μCT reconstructed 3D volumes. The seasonal porosity signal was much clearer in the deeper sample (HP1). It was concluded that the poor seasonal porosity signal in W1 was due to 1) higher instability of oceanic parameters (i.e., SST, insolation, wave energy) at shallower water depths and 2) slower extension rates which dampened the growth banding signal leading to poor clarity in seasonal growth signals. The clearer signal in HP1 lead to its use hereafter to understand seasonal variation in TD.

The method of core extraction is explained by Vincent & Sheldrake[41]. A brushed cement plug was inserted into the core hole after extraction to allow the coral to recolonise. The sample was subsequently bathed in a bleach solution (one part 7 % active chlorine bleach to three parts water) for 62 h to remove organic material. The core was thoroughly rinsed with fresh water and left to dry at room temperature for 48 h before being exported following the CITES protocol to the Department of Earth Sciences, University of Geneva. The core was cut carefully along the major growth axis using a diamond tipped saw, assuring that the cutting plane followed the growth direction of the corallites to avoid growth axis related temporal distortion during reconstructions. One half of the sample was used in this study and was sectioned again to remove the curvature of the core. The resulting slab was approximately 66 mm in length and 14 mm thick. The slab was submerged in deionised water and ultrasonically cleaned for a total of 20 min at 5-min intervals to remove cutting debris. The bath water was changed at the end of each interval. The sample was left to dry at 45 °C for 72 h and was stored in screw-top containers at room temperature before μCT-scanning and LA-ICP-MS analyses. Staining from tissue was evident by an orange band at the growth surface (see Supplementary Fig. S.1). The thickness of this band was measured using a calliper (7.17 mm). Prior to LA-ICP-MS analyses, the surface of the sample underwent a two-step polishing process: 1) diamond grinding plate to expose a single corallite tract and 2) silicon carbide polishing paper (1200/4000) to provide a flat, polished surface for ablation. The slab was inserted into deionised water and ultrasonic bathed 5 times for 5 min, changing the solution between sonification intervals to remove debris from the grinding and polishing process. The slab was left to dry before LA-ICP-MS analyses at 40 °C for a minimum time of 72 h.

## Micro CT-scanning and pixel segmentation—reconstructing porosity
Micro-CT scanning (μCT) was conducted at the Haute École du Paysage, d'Ingénierie et d'Architecture (HÉPIA) in Geneva, Switzerland. The coral slab was scanned in a (low density) polystyrene mount and vertically positioned securely on a rotating base plate to ensure that the x-ray tube aligned perpendicular to the major growth axis of the coral. The X-ray emission current was 61 mA with a voltage of 130 kW. The X-rays were directed through and around the slab before being collected on a two-dimensional X-ray detector. This detector produces a projection image/radiograph containing pixels of varying greyscale values. This process was repeated multiple times whilst the sample was rotated helically, producing 4320, two-dimensional radiographs (16-bit TIFF images) of HP1. Using the volume graphic software "MyVGL" the three-dimensional slab was reconstructed perpendicular to the major growth axis (i.e., following corallite tracts), with a total of 2254 y-axis slices at a voxel resolution of 29.2 μm. The voxel resolution was limited by the dimensions of the sample.

The slices were processed individually using a two-step image segmentation method using the R programming language[85], as described by Vincent & Sheldrake[41]. This method uses carefully reconstructed y-axis slices which are pedicular to the major growth axis and parallel to the growth bands. The first step is instance segmentation of corallites followed by semantic segmentation of individual pixels into either pores or skeleton. This allows pores and skeleton within the corallite to be classified and thus porosity to be calculated, expressed as percentage.

## LA-ICP-MS analyses
Laser ablation inductively coupled plasma mass spectrometry was used to measure $^7$Li, $^{25}$Mg, and $^{137}$Ba along the major growth axis of HP1-2 at the Department of Earth Sciences, University of Geneva. $^{43}$Ca was used for internal standardisation. A total of 1200 spots were analysed on four independent theca structures on HP1-2 over four analytical sessions (see Supplementary Fig S.5). Spots in transect 1 (T1) were spaced at 360-micron intervals. Transects 2, 3 and 4 (T2, T3, T4) were spaced at 180-micron intervals. Each spot in each analytical session was ablated using a pulsed 193 nm ArF excimer laser with an energy fluence between 4.5 and 5.5 J/cm² and a repetition rate of 10 Hz. The laser beam was circular with a diameter of 60 μm.

Counts of $^7$Li, $^{25}$Mg, and $^{137}$Ba were collected using an Agilent 8900 Triple Quadrupole ICP-MS in single quad mode. The helium flow rate was set to 800 ml/min at the start of each analytical session and was subsequently increased to 850 ml/min halfway through the session to increase ICP-MS sensitivities. An external reference material NIST612 glass standard was measured before and after each analytical block to calibrate signal intensities to their known concentrations and to correct for instrumental drift. Additionally, a matrix-matched JCp-1-NP carbonate nano powdered reference material was measured at the start of each analytical block for quality control[86].

The measured element ratios were filtered using the limits of detection (LoD) which were calculated using equation below from Longerich et al.[87]:

$$LoD = 3\sigma_{sd}\sqrt{\frac{\frac{1}{N_{bg}} + \frac{1}{N_{an}}}{S}}, \quad (1)$$

where $\sigma_{sd}$ represents the standard deviation of the background of each acquisition, $N_{bg}$ the number of counts in the background, $N_{an}$ the number of counts in the analyte signal, and $S$ the sensitivity. The sensitivity gives the net count rate obtained for an analyte per concentration unit and was calculated using the equation below from Longerich et al. (1996)[87]:

$$S = \frac{R_{N_{cal}}}{C_{N_{cal}}}\left(\frac{R_{is_{sam}}}{R_{is_{cal}}}\frac{C_{is_{cal}}}{C_{is_{sam}}}\right). \quad (2)$$

$R_{N_{cal}}$ is the count rate of the analyte in the calibration material; $C_{N_{cal}}$ is the concentration of the analyte in the calibration material; $R_{is_{sam}}$ is the count rate of the internal standard in the sample; $R_{is_{cal}}$ is the count rate of the internal; $C_{is_{cal}}$ is the concentration of the internal standard in the calibration

material; $C_{is_{sam}}$ is the concentration on the internal standard in the sample. Outliers were removed by identifying data points that fell outside 1.5 times the interquartile range (IQR), specifically below Q1 (25th percentile)—1.5 * IQR and above Q3 (75th percentile) + 1.5 * IQR.

The analytical accuracy and precision for all transects were calculated using the JCp-1-NP measurements from each analytical session ($n = 36$) and can be found in the Supplementary Information Tables 1–5). The analytical accuracy for Li/Ca, Mg/Ca, and Ba/Ca were calculated using the official reference values[88] and were 115.8 %, 107.8 %, and 113.4 % respectively. The deviations from the reference values are consistent with the limited number of published data for JCp-1-NP[88–94].

The precision of the JCp-1-NP measurements is expressed as two times the relative standard deviation (2RSD) and was calculated for each analytical session (see Supplementary Information Tables 1–5). We report precision values of 11.2 %, 8.3 %, and 8.5 %, for Li/Ca, Mg/Ca, and Ba/Ca respectively ($n = 36$). Despite the limited number of published values for JCp-1-NP, the measured precision in this manuscript falls within the reported range for LA-ICP-MS measurements, which typically have and RSD of less than 15 %[37,88–91,93,95]. Jochum et al. (2019)[91] stated that the precision of an analyte in natural reference materials depends on their concentration. Elements with relatively higher concentrations have higher precisions. Our measurement precision aligns with published values and either reflects 1) the natural heterogeneity of the *Porites* reference material and/or 2) the relatively low concentrations of $^7Li$ and $^{137}Ba$ in the JCp-1-NP reference material which influences the counting statistics[93]. The precision thus reflects the reference materials natural compositional variability, not only the LA-ICP-MS performance. As such, we believe the measurements are reliable.

To spatially synchronise the porosity reconstructions with the element ratios, the x- y- coordinates of each ablated spot were measured using stitched images from a Keyence VHX-7000 digital microscope. The slope between the first and last spot was calculated (i.e., the hypotenuse) and the total distance was projected onto a common x-axis (i.e., the adjacent) which accounted for variations in the slope angle between points. To correct for offsets between the reference image (see Supplementary Fig. S.5) and the tomography slices, the sides of the sample were aligned in both images and the angle between the images adjusted for. The slice corresponding to the starting point of the laser profile was chosen between the slice with the first reconstructed skeleton and fully reconstructed corallites (slice #2193).

A localised polynomial regression (LOESS) with a span distance of 6486 µm was applied to the geochemical data to predict the elemental ratios of the sample at the same spatial resolution as the micro-CT data (i.e., 29 µm). This approach preserved local trends in the data while reducing high-frequency noise. The span distance was chosen at a near-annual extension rate (5.89 mm/yr) to capture seasonality without overfitting to minor fluctuations. The fitted LOESS model ensured that both geochemical and porosity datasets could be directly compared on the same spatial scale for integrated analyses. It was not necessary to apply a LOESS to the porosity data because the seasonality was already clear.

## Measuring the full width at half maximum

The traditional method of calculating annual extension rates is by measuring the linear distance between high density bands (i.e., low porosity bands) which corresponds to one annual cycle of coral growth[96]. In this study however, we reconstruct the seasonal extension rates by measuring the full width at half the maximum (FWHM) of each peak and trough in the corallite porosity reconstruction and elemental ratios.

Each porosity trough was isolated by the position of its bracketing peaks. The FWHM was measured on the fitted Gaussian function of the troughs, once any difference between the bracketing peaks values was detrended and removed. This yielded the position of the lowest value in the curve (i.e., the mid-point), the FWHM (i.e. seasonal extension rate), and the mean porosity above the FWHM (i.e., seasonal porosities). Based on the difference between the positions of the troughs, annual linear extension rates (µm/year) were also calculated. Due to the bimodal nature of the porosity

peaks, it was necessary to measure the FWHM non-parametrically. Each peak was isolated based on the locations of the troughs, which were constrained using the method described above. The FWHM of the peaks were determined by first halving the distance between the bracketing troughs to define the mid-point. The peak intensity was defined by using the 95th percentile, and the half maximum threshold was set to half this value. Data points exceeding this threshold were identified, and their mean was calculated to represent seasonal porosities. The FWHM was then calculated as the difference between the maximum and minimum distance value within this subset.

The spatial uncertainty of the peak and troughs positions in the geochemistry cycles are influenced by 1) the natural variability of the signal within the sample[56] and 2) the analytical error (i.e., the precision of the JCp-1-NP reference material). The latter of which affects the LOESS smoothing and, in turn, the spatial position of each geochemical cycle. These spatial positions are used to calculate the offset between porosity and Li/Mg and Ba/Ca cycles which we interpret as past TD. Seeing as each cycle has a different extension rate (i.e., wavelength), the effective spatial resolution of the laser transects are different for each cycle. The analytical error of each session varies (see Supplementary Information ST. 1–5) which will also influences the spatial uncertainty of the cycles.

To assess the influence of natural signal variability within the sample[56], analytical sessions were combined in different configurations and at varying spatial resolutions. Cross correlations and lag distances were calculated for each configuration with the median value of all datapoints. To evaluate the influence of analytical error on our measurements, we applied a bootstrapping approach ($n = 50,000$) to the complete dataset of 1200 points collected across all analytical sessions (grey dots in Fig. 2). The session-specific 2RSD of the measurements were applied to the bootstrapping so that each iteration resampled each measurement within the 2RSD range. A LOESS smoothing (span of 0.1) was performed at the end of each iteration (i.e., resampling) to assess how the propagated analytical error (i.e., the y-axis error) for each point affected the smoothed results and thus the variability in the positions of the observed peaks and troughs (i.e., the x-axis error).

The median value of the total, spatially synchronised, bootstrapped data ($n = 1200$) was used for the geochemical FWHM measurements. A Gaussian function was fitted to each peak and troughs of the Li/Mg and Ba/Ca profiles to measure the FWHM of each cycle. The middle-point of the FWHM was subsequently used to measure the offsets between the geochemical and porosity cycles which we interpret as TD.

## Coral bleaching HotSpots

The Coral Bleaching HotSpot value is a metric developed by NOAA Coral Reef Watch (CRW) program based on work from Goreau & Haynes[97] and Montgomery & Strong[98] to measure the occurrence and magnitude of instantaneous heat stress that causes coral bleaching (see NOAA CRW for details). HotSpot values are defined in specific locations where the SST increases above the maximum monthly mean SST. This calculation was made on the 90th percentile of SST data accessed from the NOAA Satellite and Information Service (NESDIS) under the NOAA CRW program. The 5 km Regional Virtual Stations (version 3.1) provided daily SST data for the windward Caribbean islands (polygon middle longitude: −60.4750 and latitude: 13.4000) spanning from 1985 to 2024. Data was extracted covering 2013 to 2024 which corresponds to the reconstructed timespan based on the reconstructed growth banding. The data plotted in Fig. 4 corresponds to HotSpot values between September to December which coincide with the annual peak in SST during the W/W-season.

## Data availability

Pixel segmented micro-CT data, LA-ICP-MS data, degree hotspot values from NOA CRW and precipitation data from Grantley Adams Airport, Barbados, can be accessed here: https://doi.org/10.6084/m9.figshare.30777779.

**Article**

## Code availability

All analyses were performed using RStudio version 2024.04.2 + 764 and codes are available upon request to either james.vincent@unige.ch or thomas.sheldrake@unige.ch.

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

## Acknowledgements
We acknowledge the use of the micro-CT scanner provided by the Haute École du Paysage, d'Ingénierie et d'Architecture and thank Petr Kisselev for his assistance with scanning the samples. We thank Michael Schirra and Alexandra Tsay for their assistance setting up the LA-ICP-MS. We thank the technicians at the University of Geneva for their help with sample preparation (Jean-Marie Boccard, Fréderic Arlaud and François Gischig). We acknowledge the Coastal Zone Management Unit in Barbados for their support. Coral sampling was conducted under the permission of the Coastal Zone Management Unit. Samples were exported from Barbados and imported to Switzerland with CITES permit number 24EB000707-AS. We would like to thank the four anonymous reviewers who's comments significantly improved this manuscript. This research was supported by the Swiss National Science Foundation under grant number 194204.

## Author contributions
James Vincent and Tom Sheldrake contributed to all aspects of this work.

## Competing interests
The authors declare no competing interests.
