## [Transparent Peer Review file · Communications Earth & Environment]

Coral tissue depth reconstructed using skeletal microstructural offsets is driven by environmental stress

Corresponding Author: Mr James Vincent

Version 0:

Decision Letter:

Dear Mr Vincent,

Your manuscript titled "Stress-limited soft tissue thickness drives future calcification in corals" has now been seen by 3 reviewers, whose comments are appended below. You will see that they find your work of some potential interest. However, they have raised quite substantial concerns that must be addressed. In light of these comments, we cannot accept the manuscript for publication, but would be interested in considering a revised version that fully addresses these serious concerns.

In particular, we would like to ask you to address the concerns regarding the robustness of the proposed approach:

- (a) Consider the possibility for additional data collection (i.e., increase number of replicate measurements, consider analysing the second available coral core from your previous study),
- (b) Present compelling evidence or arguments regarding the concerns of the quality of the geochemical data.

We hope you will find the reviewers' comments useful as you decide how to proceed. Should additional work allow you to address these criticisms, we would be happy to look at a substantially revised manuscript. If you choose to take up this option, please either highlight all changes in the manuscript text file, or provide a list of the changes to the manuscript with your responses to the reviewers.

When resubmitting, please provide a point-by-point response to the reviewers' comments. Please submit your responses as a separate file, distinct from your cover letter where you can add responses to the Editors' comments that you do not want to be made available to the reviewers. Word files are preferred. We recommend that any figures, tables or graphs that are included in the response to reviewers are also included in the main article or Supplementary Information.

If the revision process takes significantly longer than three months, we will be happy to reconsider your paper at a later date, as long as nothing similar has been accepted for publication at Communications Earth & Environment or published elsewhere in the meantime.

Please use the following link to submit your revised manuscript, point-by-point response to the reviewers' comments with a list of your changes to the manuscript text (which should be in a separate document to any cover letter), a tracked-changes version of the manuscript (as a PDF file) and any completed checklist:

Link Redacted

Please do not hesitate to contact us if you have any questions or would like to discuss the required revisions further. Thank you for the opportunity to review your work.

Best regards,

Nadine Schubert, PhD
Editorial Board Member
Communications Earth & Environment
orcid.org/0000-0001-7161-7882

Alice Drinkwater, PhD
Associate Editor
Communications Earth & Environment
Consulting Editor
Communications Sustainability

EDITORIAL POLICIES AND FORMAT

If you decide to resubmit your paper, please ensure that your manuscript complies with our editorial policies and complete and upload the checklist below as a Related Manuscript file type with the revised article:

Editorial Policy [Policy requirements](https://www.nature.com/documents/nr-editorial-policy-checklist.pdf) (Download the link to your computer as a PDF.)

- Behavioural and social science
- Ecological, evolutionary & environmental sciences
- Life sciences

<https://www.nature.com/documents/nr-reporting-summary.zip>

For your information, you can find some guidance regarding format requirements summarized on the following checklist: (<https://www.nature.com/documents/commsj-phys-style-formatting-checklist-article.pdf>) and formatting guide (<https://www.nature.com/documents/commsj-phys-style-formatting-guide-accept.pdf>).

REVIEWER COMMENTS:

Reviewer #1 (Remarks to the Author):

Vincent and Sheldrake present a novel and interesting approach by combining coral geochemical and growth data to infer metabolic changes related to coral health. Their work highlights the broader potential of such data beyond traditional environmental reconstructions and demonstrates the value of a multiproxy strategy. Their findings suggest that current thermal stress levels, along with critically low coral energy reserves, are compromising not only coral growth but likely other essential physiological functions as well.

While I commend the innovation of their approach, I found the manuscript challenging to follow. Despite my familiarity with the subject, I needed to read the text multiple times to grasp the underlying logic. I strongly recommend that the authors more clearly and systematically explain the key concept of the offset between calcification, the geochemical signal, and how this offset translates into inferences about tissue thickness.

Moreover, the concept of mass accumulation and "biosmoothing" first introduced by Gagan et al. (2012) is highly relevant to this study but is surprisingly not mentioned. If the authors' inferences are correct, then the degree of biosmoothing in the geochemical signal should correlate with reconstructed changes in tissue thickness. I would expect that the residuals between the geochemical signal and actual temperature records could provide further insight: as extension rates slow, increased overprinting and smoothing should occur.

Also, it is known that the timing of density band formation can vary between individuals in *Porites* corals (e.g., Barnes and Lough, 1992). Could a similar variability occur in *Siderastrea*? If so, how might it affect their interpretations?

Specific comments:

- Line 8: Are reproduction and mucus production included in the energy allocation framework? These are major energy-consuming processes and are only briefly mentioned later.
- Line 31: While this represents the general model, it is not universally applicable. For instance, see Barnes and Lough

(1992).

- Line 42: The concept of dissepiment formation requires context. Additionally, dissepiment identification can be challenging in species where their structure is less obvious.
- Line 48: Is energy allocation the central objective of the study, or is it an indirect interpretation? At no point are these aspects directly measured; rather, the study infers them.
- Line 56: FWHM (Full Width at Half Maximum) appears for the first time here and should be properly introduced.
- Line 69: Shouldn't both datasets (geochemical and growth) be smoothed in a consistent manner?
- Line 75: The text assumes the reader is aware that terrestrial runoff is seasonal and peaks during summer at the study site. This should be explicitly stated.
- Line 78: "Compared to the Ba/Ca?" — this comparison needs clarification.
- Line 171: Can corals truly digest volcanic ash? Clarification or references would be helpful.
- Figure 1: It might be clearer to present the red and blue areas explicitly. Also, shouldn't the labels on the top x-axis be "W-S" (winter-summer)?
- Line 339: Similarly, "W-S" should be mentioned here for consistency.
- Supplementary Figure 1:
 - o Was the relationship between the reconstructed soft tissue thickness and actual tissue thickness confirmed through measurements?
 - o How was it determined that the interval between the pink (ii) and yellow (iii) stars represents the soft tissue depth where pre-existing skeleton thickening occurs? This is a central concept and requires more detailed explanation.
 - o The reconstructed TSS does not match the tissue thickness data from Vincent and Sheldrake (2025), how you reconcile this?
 - o Please clarify that panel e refers to the calyx depth in *Siderastrea*. How confident are the authors that the measured structure corresponds to the calyx depth?
 - o The assumption of a V-shaped calyx may not always hold: in some *Porites* species, the columella and adjacent pali can create a W-shaped profile, potentially increasing actual calyx depth relative to a simple V-shape.

Reviewer #2 (Remarks to the Author):

Vincent and Sheldrake examine how massive scleractinian corals allocate energy between skeletal growth and soft tissue formation, revealing that seasonal tissue thickness is influenced by environmental stress. The authors suggest that variations in soft tissue thickness directly drive skeletal extension and that offsets between geochemical cycles in the skeleton and growth bands may provide a means to reconstruct soft tissue thickness. The study finds that coral bleaching HotSpot values, which indicate climate change-related stress, are inversely correlated with growth band offsets, affecting future calcification. Additionally, the authors show that major environmental events, e.g. volcanic eruptions, may play a role in coral stress responses, offering insights into how corals have historically adapted to changing conditions.

This is an interesting paper on an important topic. The authors use a new method introducing a potential coral stress response indicator and test this novel indicator against recent warm/stress events. The findings are a very valuable contribution to the field and are of major interest to the community. Thus, the work is important and worthy of publication in a journal like *Communications Earth & Environment*. The conclusions are convincing, however there are some ways that the manuscript can be improved to strengthen the conclusions, and thus I would recommend that the authors revise the manuscript. Detailed comments are given below.

Detailed comments:

The authors reference their previously published paper from the same study site (line 60), which has focused on micro-CT analysis and the establishing of their method to analyze growth banding and classify microstructures and reconstructing skeletal porosity. This study has used the same core from 15 m water depth (referenced in Suppl. Fig.1 and reference number 53), plus a coral core from 5 m water depth at a second site. I am wondering why here only one of the cores is used for LA-ICP-MS analysis (15 m water depth core from NW-Barbados). If laser data is not available for the shallower site, it would be helpful to add it or give a reason why the core from the deeper site has been chosen for this study. The authors suggest that their results could be very helpful for predicting coral response to anthropogenic stressors (such as high SST events). Thus, having data from shallower water, where water temperature is often showing higher variability (and more likely to be affected by temperature extremes as regularly captured by satellite SST observations) would be useful.

The Amazon and Orinoco Rivers significantly influence the study site due to its position at the southeastern margin of the Caribbean Sea. This is briefly noted in the Methods (line 463). Owing to this geographic setting, a pronounced seasonal influence on Barbados is expected as North Brazilian Current rings episodically transport waters originating from these rivers into the region. These riverine plumes are characterized by elevated sediment and nutrient loads, which may reach the study site during certain periods of the year. Is the coral Ba/Ca ratio potentially influenced by this transport of sediment-loaded waters, which also bring nutrients to the study site? Given the observed clear and recurring seasonal cyclicity in the coral Ba/Ca ratios, it is plausible that these ratios are modulated by the periodic arrival of sediment- and nutrient-rich waters from the Amazon and Orinoco Rivers. Alternatively, local processes that deliver Ba-enriched waters to the site could also contribute to the seasonal Ba/Ca signal. However, the timing and nature of the Ba/Ca variability are consistent with the established seasonal dispersal patterns of the Amazon and Orinoco River plumes in the eastern Caribbean. Therefore, I suggest to discuss the potential contribution of Amazon and Orinoco River discharge as a source of barium in greater detail within the main text, especially with respect to the timing of the seasonal cycle when comparing potential local vs. more distant sources of sediment/Ba input.

minor comments:

line 337 - Figure 1: raw Ba/Ca and Li/Mg ratios (thin solid lines) look fine in the pdf but not bright enough when printed on paper.

line 90: 3- to 4 year cycles are reported in the offsets, which are linked to timing of rainfall and SST at the site. Earlier studies have shown potential links to El Nino Southern Oscillation in records from the Caribbean Sea in this frequency band.

Reviewer #3 (Remarks to the Author):

General comment

The study proposed a novel methodology for measuring past tissue layers on coral records. This approach combines microstructure and geochemical data over a 60 mm segment of a single coral core from the species *Siderastrea siderea*, collected from the coast of Barbados in the Caribbean Sea. The authors claim that this approach can be applied through a coral core to produce information about past tissue layers conditions and how it is influenced by environmental stressors on the coral. While the approach is intriguing and offers novel insights into paleoclimate and environmental reconstruction based on coral archives, it is essential that the necessary data is thoroughly tested before its implementation.

Major considerations

The primary concern with this approach is its lack of robustness, due to its basis on a single geochemical and microstructure record from a single colony. I am aware that geochemical measurements can result in significant costs, however, for a study proposing a novel approach that could be of considerable benefit to the scientific community, I believe it would be advisable for the authors to collect additional data. As the authors are working with small sections of coral cores, replication is very feasible; at least 5 to 10 replicas could be obtained. Even though new field work for the collection of additional coral cores can be complex, it is still possible to replicate the analyses using the same coral slab. For example, they could provide five geochemical transects along five parallel theca walls concomitantly with porosity measurement on the same coral slab. Additionally, reference number 53 (Vincent, J. & Shel Drake 2025), which is from the same authors, presents micro CT-Scan data on two *S. siderea* corals. Why didn't they explore at least these two records? The authors could even get 3 to 5 replicas on geochemical and porosity data from each coral core to increase the robustness of their interpretation. From my own experience of replicating geochemical measurement on different theca of the same coral slab, I know that slight variations in results can be expected for a number of reasons. This is a key consideration in this new approach. The primary question to be addressed by this study is to determine whether different paired geochemical-porosity records yield the same relationship. If so, we can be confident that this new approach can be used to reconstruct previous tissue layers and infer possible stressors from different records of *S. siderea*.

A second and also major issue is the quality of the geochemical data produced by LA-ICP-MS, which is known to be noisy. The authors claimed that the high RSD wouldn't impose a limitation on the interpretation of geochemical cycle. I have some concerns about that, which is aggravated by the lack of replications. A noisy record could drift the smooth line when assuming the highest and lowest values of the record. That can compromise the whole history raised by the authors. A further question to be addressed is that of the rationale behind the authors' decision to not utilize Sr/Ca ratios as a proxy for SST, given the long- and well-established nature of this method. The selection of Li/Ma and Ba/Ca ratios was predicated on which criteria?

A personal recommendation to the authors is to consider adopting a more conventional approach, such as micromilling techniques for producing powder that can easily dissolved in HNO₃ and analyzed by ICP-MS, which will probably result in much more precise geochemical data. The coral HP1 exhibits a growth rate that permits the generation of more than 12 samples per year through micromilling, by using a sampling resolution of 0.5 mm. This resolution is sufficient to report the full geochemical seasonality of the coral (see the work of Maupin et al., 2008, <http://dx.doi.org/10.1029/2008GC002106>). I believe that would increase the robustness of the geochemical data of this study. The author should also mention the growth rate of the colony, they have this information.

The present study, which proposes a novel methodology for the assessment of past tissue thickness, lacks a thorough examination of the aforementioned issues. Consequently, it is not sufficiently robust to be published in Nature Communications Earth & Environment.

Additional Important points:

1. Please, provide a figure with the coral core slab image and the location where the geochemical and porosity data were acquired. This is very important for what you are proposing.
2. The AIMS the study "how energy is allocated during seasonal growth" and "how stress influences energy allocation", it is not appropriate for the study. The authors should first establish the approach and not dive into more complex questions like those one proposed.
3. The authors should address properly why they observed 10 geochemical cycles against 9 porosity cycles.

4. Li/Mg and Ba/Ca are mostly in phase through the record. First, I would compare this with the local sea surface temperature (SST) and precipitation to check if the pattern is also happening in real environmental conditions. I would also make a crossplot of the Li/Mg and Ba/Ca geochemical data, as well as a crossplot of the SST and precipitation data to check their covariance. I would also suggest that the authors include a figure showing the climatology of SST and precipitation for the location, since they are assuming that the porosity index, Ba/Ca and Li/Mg are related to these variables. This would help them understand the index better. There are available data on precipitation and SST for the region and a time series from 2022 to 2012 would make the work much better.

5. Why did the study suddenly change to focus on the volcanic event in Barbados? You are still trying to work out the approach! Moreover, there's a lot of speculation about this topic, which could only be addressed by investigating longer coral records that registered the volcanic event in 1979.

Specific comments:

Lines 27-31. As the manuscript focus in coral skeletal structure I think some very important citations are missing here. It would be nice to cite the first discoveries on the topic, for example the works of:

Knutson et al 1972 [<http://dx.doi.org/10.1126/science.177.4045.270>]
Buddemeier et al 1974 [[http://dx.doi.org/10.1016/0022-0981\(74\)90024-0](http://dx.doi.org/10.1016/0022-0981(74)90024-0)]

Line 57. Define FWHM at your first entrance in the text, readers might not be familiar with the term.

Lines 337-347. The caption of figure 1 is hard to follow, I suggest reformulating the figure for a better visualization of what the authors want to say. The blue and red dash lines are confusion. The authors presented the porosity record twice and I don't see a practical reason. It must have a better way to show the data. Moreover, they mentioned a vertical "grey dashed line" which is not present in the figure.

Line 74: The authors should specify which extreme they are mentioning. Are they mentioning the first highest value? If so, I suggest to insert a text in the Fig.1 A and B informing that this first value is interpreted as July 2022, that is why I am supposing that the core was collected in 2022, there is no information of the year of collection in this manuscript.

76: Again, the term extreme without specification. I suggest the author replace this term by "lowest or highest peak".

77: should be "Fig. 1a"

78: there is no "Fig. 1c"

Lines 80-84: That is truly hard to sell. The text is confusing, for example, the scale they use in the figure did not help the readers visualize where the Li/Mg peak is, which is even more complicated because there are one peak from the smooth and another more positive value from the raw data. The previous issue is simple to fix. The main problem is that they observed that the distance between the most recent porosity trough and Li/Mg peak matched the microstructure of what they call "thickening". However, this cannot be taken as absolute truth – it needs to be tested again.

I suggest that the authors adds the supplementary figure 1 to the main text figure 1 and adds it as a zoom detail at the top of the record. That would make it much easier to understand their reasoning. It's tiring going back and forth to check the figure while reading the text.

Authors should improve the provide and age model for figure 1 at the top of the figure. That will help to clean up the excessive labels at the top of the figure.

Lines 152-186. That is a very interesting topic, but highly speculative. The authors should drop this topic and not include it in the current manuscript. It should be left for another assessment where replications are properly done and longer records are available, which include another volcanic event.

Communications Earth & Environment is committed to improving transparency in authorship. As part of our efforts in this direction, we are now requesting that all authors identified as 'corresponding author' create and link their Open Researcher and Contributor Identifier (ORCID) with their account on the Manuscript Tracking System prior to acceptance. ORCID helps the scientific community achieve unambiguous attribution of all scholarly contributions. You can create and link your ORCID from the home page of the Manuscript Tracking System by clicking on 'Modify my Springer Nature account' and following the

instructions in the link below. Please also inform all co-authors that they can add their ORCID to their accounts and that they must do so prior to acceptance.

Version 1:

Decision Letter:

Dear Mr Vincent,

Your revised manuscript titled "Reconstructing environmental stress in corals by quantifying tissue depth using skeletal microstructural offsets" has now been seen again by the 3 reviewers, whose comments are appended below. In the light of their advice we regret to inform you that we cannot publish your manuscript in Communications Earth & Environment.

You will see that 2 of the reviewers raise ongoing substantive concerns around the interpretation of the data in support of your central claim. Taking these points together with our editorial considerations, we are unable to conclude that your manuscript represents a sufficiently novel and compelling advance over the body of related work in the literature. Unfortunately, these reservations are sufficiently important to preclude publication of this study in Communications Earth & Environment.

As a service to our authors, we have also assessed potential suitability of a revised version of your article for consideration in Communications Sustainability, a new journal from the Nature Portfolio.

We are sorry that we cannot be more positive on this occasion and thank you for the opportunity to consider your work.

Best regards,

Nadine Schubert, PhD
Editorial Board Member
Communications Earth & Environment
orcid.org/0000-0001-7161-7882

Alice Drinkwater, PhD
Associate Editor
Communications Earth & Environment
Consulting Editor
Communications Sustainability

Reviewers' comments:

Reviewer #1 (Remarks to the Author):

General comments

Vincent and Sheldrake have done a very in-depth review of the manuscript, and I commend their thorough attempt to address all reviewer comments. However, I feel that some of the implemented changes have not improved the manuscript in the best possible way, particularly in the Introduction.

The Introduction remains too long and unfocused. It mixes background information, methods, results, and discussion, and thus reads more like a mini-review combined with a results preview. The expected logical structure: background, knowledge gap, and hypothesis/aim is not clearly evident. Instead, the text oscillates between general background, detailed methodology, and even findings. Several paragraphs seem to belong more appropriately in the Methods or Results sections.

I am also concerned about the offset data, which no longer appears to track the hotspot data as clearly as before. This change weakens the central narrative, as it is less obvious that the data support the proposed mechanism. A scatter plot or correlation analysis might help clarify this relationship.

Specific comments

Line 8 – Please avoid the use of “/”.

Line 16 – Suggest replacing future with subsequent.

Line 36 – Generally true, though some exceptions have been reported (see: <https://www.ingentaconnect.com/content/umrmsas/bullmar/2000/00000066/00000001/art00020>).

Line 50 – The main issue here is whether tissue thickness is variable.

Line 52 – This line could be removed.

Line 61 – Paragraph feels too detailed; reads more like methodology/results.

Line 83 – Suggest removing “negative”.

Line 87 – The relative impact of volcanic eruptions is likely minor among reef stressors.

Line 110 – TD not defined.

Line 128 – This section feels more appropriate for Methods or Supplementary Information.

Line 141 – Could be condensed.

Line 166 – Suggest adding “coral”.

Line 186 – The note “(note the Li/Mg is reversed for better comparability)” is redundant; already stated in the figure caption.

Line 200 – Most of this information is only needed in the figure caption.

Line 206 – Is the offset consistent when converted to the time domain? Does the offset in SST and rainfall align with that in Li/Mg and Ba/Ca?

Line 214 – What could this represent, then?

Line 232 – Please avoid using “This” alone as a subject (“This variation...” should be specified).

Line 256 – The pattern is not evident for all years; 2013 and 2016 yes, but not others. Consider quantifying the relationship (e.g., scatter plot between thermal stress and offset).

Line 279 – In the original version, the offset decreased after 2020, whereas it now increases, yet the explanation remains unchanged. This inconsistency undermines the argument — how can the same explanation apply if the trend is reversed?

Line 321 – Please rewrite for clarity; currently hard to follow.

Reviewer #2 (Remarks to the Author):

I have no further comments and accept the manuscript in its revised form after carefully evaluating the authors response.

Reviewer #3 (Remarks to the Author):

The authors have improved the manuscript by doing three additional LA-ICP-MS transects on HP1 core. This represents an improvement; however, I still find the interpretation to be somewhat speculative. In my view, proposing a new proxy based on a single record, especially when combining multiple analytical approaches, warrants greater caution.

I acknowledge the authors' efforts to make the study as robust as possible by addressing all the comments the reviewers have made. Nevertheless, there appears to be a significant conceptual error regarding the Li/Mg proxy that could compromise their interpretation. Specifically, they state in lines 183–186:

“Given the sample was collected in July 2022, transitioning from C/D-season to W/W-season, we interpret the closest porosity trough to the growth surface (orange star – Fig.3) to represent the C/D-season with low SST's when investment of energy into linear skeletal extension is expected to be lower.”

However, the peak corresponding to lower Li/Mg values (orange star) indicates higher SST. How, then, do the authors interpret this as representing a cold/dry season? Low Li/Mg ratios thermodynamically correspond to higher temperatures, as demonstrated in several studies, including Cuny-Guirriec et al. (2019), which the authors cite, rather than the opposite.

Again, in lines 189 to 193

“On this basis, high porosity bands correspond to the W/W-seasons characterised by fast skeletal extension rates (ranging between 3.56 to 5.17 mm/season), whilst low porosity bands correspond to C/D-seasons with slower extension rates varying

between 1.25 and 2.17 192 mm/season (Fig.4).”

That is not what the Li/Mg data indicates. They show higher Li/Mg ratios, which correspond to colder SST.

And again, in lines 200-204

“The position on the porosity troughs (grey stars – Fig. 3) are projected into the Li/Mg and Ba/Ca profiles (grey dashed lines – Fig. 3) to highlight these offsets between the porosity troughs and corresponding Li/Mg peaks and Ba/Ca troughs which are hereafter termed C/D-season offsets (note that y-axis is reversed for Li/Mg in Fig. 3). The porosity peaks show a similar negative offset with Li/Mg and Ba/Ca cycles which is hereafter termed W/W-season offset.”

The grey lines are not indicating Ba/Ca thoughts!

This is not a minor oversight. The misinterpretation affects all subsequent sections that rely on the Cold/Dryer and Warm/Wetter assessments, which at some point compromising the overall narrative of the TD proxy application.

Version 2:

Decision Letter:

Dear Mr Vincent,

Your manuscript titled "Reconstructing environmental stress in corals by quantifying tissue depth using skeletal microstructural offsets" has now been seen by our reviewer, whose comments are appended below. In the light of their advice we regret to inform you that we cannot publish your manuscript in Communications Earth & Environment.

You will see that the reviewer raises substantive concerns around low replication and limited dataset in the study. Taking these points together with our editorial considerations, we are unable to conclude that your manuscript represents a sufficiently novel and compelling advance over the body of related work in the literature. Unfortunately, these reservations are sufficiently important to preclude publication of this study in Communications Earth & Environment.

As a service to our authors, we have also assessed potential suitability of a revised version of your article for consideration in Communications Sustainability, a new journal from the Nature Portfolio.

We are sorry that we cannot be more positive on this occasion and thank you for the opportunity to consider your work.

Best regards,

Nadine Schubert, PhD
Editorial Board Member
Communications Earth & Environment
orcid.org/0000-0001-7161-7882

Alice Drinkwater, PhD
Associate Editor
Communications Earth & Environment
Consulting Editor
Communications Sustainability

Reviewers' comments:

Reviewer #4 (Remarks to the Author):

General comments:

The manuscript entitled "Reconstructing environmental stress in corals by quantifying tissue depth using skeletal microstructural offsets" explores the potential of using geochemical proxies in the *Siderastrea siderea* coral skeleton to reconstruct past tissue-layer depth and assess environmental stress on corals. This study presents an interesting and potentially valuable approach for the biogeochemical community and for researchers utilising coral geochemical proxies. However, the manuscript still contains uncertainties regarding the ability of the proposed tool/approach to effectively reconstruct past tissue-layer depth. Therefore, I recommend minor revision of the manuscript, as outlined in the detailed comments below. In addition, Figure 3 requires improvement before the manuscript can be accepted

for publication to prevent possible misunderstandings among readers.

Note:

My review focuses on two major concerns, as requested by the editor:

1. A possible conceptual error and misinterpretation of the coral skeletal proxy Li/Mg
2. The adequacy of the number of records to support the proposed tool for reconstructing past tissue depth and assessing environmental stress on corals

Concern (1) Possible conceptual error and misinterpretation of the coral skeletal proxy Li/Mg – Figure 3:

In my review, I did not identify any conceptual errors or misinterpretations of the coral Li/Mg proxy. As described in the manuscript (L41-42, 119-120), coral Li/Mg can serve as a temperature proxy. It is well established that coral Li/Mg inversely correlates with temperature (i.e., higher Li/Mg values correspond to lower temperatures) (e.g., Hathorne et al., 2013; Fowell et al., 2016).

However, Figure 3 may lead to potential confusion among readers. The authors note that the Li/Mg record is plotted in inverse orientation on the y-axis, and that the offsets are indicated by symbols (stars), grey dashed lines, and arrows in both the figure and its caption. Although L157-164 provide detailed explanations, the figure may still be misleading to some readers.

To mitigate this risk and better highlight the interesting findings, I recommend the following improvement to Figure 3:

In the current figure, the orange and grey stars indicate cold/dry seasons, and the offsets between porosity and chemical profiles are illustrated by grey dashed vertical lines and arrows. While these annotations help identify offset positions, they might cause confusion because the stars (denoting cold/dry seasons) appear only in the corallite porosity panel. I suggest that the authors also add similar stars in the Li/Mg (and Ba/Ca) panels to indicate cold/dry seasons for each cycle. This modification would more clearly demonstrate the offset relationships and the correspondence of seasonal cycles among corallite porosity, Li/Mg, and Ba/Ca records.

Concern (2) Number of records supporting the proposed "tool":

In the abstract (L18–19), the authors state that this study "provides a tool to reconstruct past tissue depth and reconstruct environmental and ecological stress." The approach presented is indeed novel and has strong potential for reconstructing past tissue depth using *Siderastrea siderea* corals. However, since the study is based on a single coral core ($n = 1$), it remains essentially a case study and does not yet justify being described as a new "tool."

Although the revised manuscript includes additional geochemical data from three transects, all transects were derived from the same coral core. These can be regarded only as within-core replicates rather than independent confirmations.

Furthermore, it remains uncertain whether the specimen (HP1) used in this study is representative of *S. siderea* colonies with V-shaped corallites in the study area. To validate the proposed approach, a multi-colony-based study would be required to statistically confirm its reproducibility and robustness. Therefore, I recommend that the authors replace the term "tool" with "approach" or "concept" throughout the manuscript (not only in the abstract but consistently across the entire text). In addition, it would strengthen the manuscript to explicitly acknowledge the need for and encourage a future multi-colony validation study to test the robustness of this approach.

Reference:

Hathorne, E. C., Felis, T., Suzuki, A., Kawahata, H., & Cabioch, G. (2013). Lithium in the aragonite skeletons of massive *Porites* corals: A new tool to reconstruct tropical sea surface temperatures. *Paleoceanography*, 28(1), 143-152.
Fowell, S. E., Sandford, K., Stewart, J. A., Castillo, K. D., Ries, J. B., & Foster, G. L. (2016). Intrareef variations in Li/Mg and Sr/Ca sea surface temperature proxies in the Caribbean reef-building coral *Siderastrea siderea*. *Paleoceanography*, 31(10), 1315-1329.

Version 3:

Decision Letter:

Dear Mr Vincent,

Your manuscript titled "Reconstructing environmental stress in corals by quantifying tissue depth using skeletal microstructural offsets" has now been discussed by our editorial team. We are delighted to say that we are happy, in principle, to publish a suitably revised version in *Communications Earth & Environment*.

We therefore invite you to revise your paper one last time to edit your manuscript to comply with our format requirements and to maximise the accessibility and therefore the impact of your work.

EDITORIAL REQUESTS:

****Please take care to match our formatting and policy requirements. We will check revised manuscript and return manuscripts that do not comply. Such requests will lead to delays. ****

SUBMISSION INFORMATION:

OPEN ACCESS:

Communications Earth & Environment is a fully open access journal. Articles are made freely accessible on publication. For further information about article processing charges, open access funding, and advice and support from Nature Portfolio, please visit <https://www.nature.com/commsenv/open-access>

Link Redacted

Best regards,

Alice Drinkwater, PhD
Associate Editor
Communications Earth & Environment
Consulting Editor
Communications Sustainability

** Visit Nature Portfolio's author and referees' website at <http://www.nature.com/authors> for information about policies, services and author benefits**

We would firstly like to thank the reviewers for their constructive comments that have improved the robustness of the manuscript and helped validate our approach.

Reviewer 1:

1) Vincent and Sheldrake present a novel and interesting approach by combining coral geochemical and growth data to infer metabolic changes related to coral health. Their work highlights the broader potential of such data beyond traditional environmental reconstructions and demonstrates the value of a multiproxy strategy. Their findings suggest that current thermal stress levels, along with critically low coral energy reserves, are compromising not only coral growth but likely other essential physiological functions as well.

2) While I commend the innovation of their approach, I found the manuscript challenging to follow. Despite my familiarity with the subject, I needed to read the text multiple times to grasp the underlying logic. I strongly recommend that the authors more clearly and systematically explain the key concept of the offset between calcification, the geochemical signal, and how this offset translates into inferences about tissue thickness.

We acknowledge that the manuscript was difficult to follow. By addressing the concerns outlined by the reviewers and by reframing the paper from energy to soft tissue depth, and its relationships with environmental stress, we hope our corrections improve the clarity and readability of our work. We have also significantly expanded and rewritten the introduction and discussion to better explain the key concepts between corallite skeletal thickening, theca geochemistry, and how the offset translates into inferences between tissue depth (see lines 60-79 and 177-227).

3) Moreover, the concept of mass accumulation and "biosmoothing" first introduced by Gagan et al. (2012) is highly relevant to this study but is surprisingly not mentioned. If the authors' inferences are correct, then the degree of biosmoothing in the geochemical signal should correlate with reconstructed changes in tissue thickness. I would expect that the residuals between the geochemical signal and actual temperature records could provide further insight: as extension rates slow, increased overprinting and smoothing should occur.

We appreciate this comment, and we acknowledge its significance for our work. We have added this subject into the introduction on lines 48-50. The residuals between the geochemical signal and actual temperature records could provide insight when considering tissue depth and bio-smoothing which we will consider in future work.

4) Also, it is known that the timing of density band formation can vary between individuals in *Porites* corals (e.g., Barnes and Lough, 1992). Could a similar variability occur in *Siderastrea*? If so, how might it affect their interpretations?

Similar variability in the timing of density band formation for *Siderastrea* has been reported⁹ (and references there-in). We used the context of the sample (i.e., collected in July 2022) to backdate and constrain the timing of growth band formation. This is explained in detail in

Vincent & Sheldrake (2025)¹ but has now been explained on lines 126-128, 183-186 and 426-428. We try to show in the manuscript that simply spatially synchronising geochemistry with growth bands can lead to false interpretations attributed to variations in the depth at which soft tissues thicken the pre-existing skeleton. This can contribute to the contradicting timing of growth band formation reported in literature.

Specific comments:

5) Line 8: Are reproduction and mucus production included in the energy allocation framework? These are major energy-consuming processes and are only briefly mentioned later.

We acknowledge that both reproduction and mucus production are energy expensive. Given we have reframed the manuscript from energy allocation to tissue depth and environmental stress, as recommended by another reviewer, we do not talk about these topics in detail in the new version of this manuscript.

6) Line 31: While this represents the general model, it is not universally applicable. For instance, see Barnes and Lough (1992).

We have stated in the new version that this is the general approach (line 35) and have cited Barnes and Lough (1992)¹⁰.

7) Line 42: The concept of dissepiment formation requires context. Additionally, dissepiment identification can be challenging in species where their structure is less obvious.

We acknowledge that in the original version of the manuscript there was no context to dissepiments. We have added context on lines 113-122 in the updated manuscript.

8) Line 48: Is energy allocation the central objective of the study, or is it an indirect interpretation? At no point are these aspects directly measured; rather, the study infers them.

Upon reflection we agree that the energy allocation muddled the story of the original manuscript and so has been removed from this version.

9) Line 56: FWHM (Full Width at Half Maximum) appears for the first time here and should be properly introduced.

This is stated in the methods section on lines 618-645 in the original version but has been added on lines 149-151 and in the methods on lines 566-585.

10) Line 69: Shouldn't both datasets (geochemical and growth) be smoothed in a consistent manner?

In the tomography data the seasonal signal is clear and thus no smoothing was necessary. The high-resolution noise within the geochemistry data meant that smoothing was necessary to reveal the seasonal signal. This has been explained on lines 558-560.

11) Line 75: The text assumes the reader is aware that terrestrial runoff is seasonal and peaks during summer at the study site. This should be explicitly stated.

We have explained that terrestrial runoff is seasonal and peaks during summer on lines 386-400 in the new manuscript.

12) Line 78: "Compared to the Ba/Ca?" — this comparison needs clarification.

The peaks and troughs in Ba/Ca cycles occur after the Li/Mg cycles (see Figure 3). This has been explained on lines 206-207. This results in a different offset for Li/Mg and Ba/Ca (Fig. 4a), with the Li/Mg most likely representing the tissue depth of active skeletal thickening, as explained on 216-228.

13) Line 171: Can corals truly digest volcanic ash? Clarification or references would be helpful.

We agree that no clear evidence exists that corals can digest volcanic ash. Nevertheless, we have video evidence of the branching coral *Stylophora pistilata* ingesting volcanic ash during aquarium experiments presented in Förster et al., 2023⁶. Furthermore, corals are also known to ingest microplastics^{11,12}. Consequently, we have changed the terminology from "digest" to "ingest" in the manuscript on lines 289-291, as we have no direct knowledge concerning the volcanic ash once it is ingested.

14) Figure 1: It might be clearer to present the red and blue areas explicitly. Also, shouldn't the labels on the top x-axis be "W-S" (winter-summer)?

We acknowledge that Figure 1 was not clear and have remade the Figure. The "W-S" in the first version of the manuscript is an abbreviation for warm-season not winter-summer. We use warm- and cold-season as it better describes the Li/Mg ratios which show cycles of warm and cold SSTs and not necessarily the typical summer and winter months. Based on the Li/Mg and Ba/Ca ratios, we have re-termed the seasons as warm/wet (low Li/Mg and high Ba/Ca ratios) and cold/dry (high Li/Mg and low Ba/Ca ratios) for better clarity.

15) Line 339: Similarly, "W-S" should be mentioned here for consistency.

See response to the comment 14.

• Supplementary Figure 1:

16) Was the relationship between the reconstructed soft tissue thickness and actual tissue thickness confirmed through measurements?

The relationship between the reconstructed tissue and actual soft tissue could not be confirmed. For such a comparison, in situ measurements would need to be taken each season over a 10-year period which was not possible. It was also not possible to use the spacing of dissepiments in the *Siderastrea* sample to reconstruct the tissue thickness, given the difficulties mentioned in comment 7. We acknowledge this as crucial outlook research to be undertaken and have added this to lines 356-358. Nevertheless, we have compared the Li/Mg offset (i.e., reconstructed soft tissue) to the thickening process at the surface, as explained on lines 211-227. The measured offset is less than the depth of tissue staining at the surface of the sample (see supplementary figure S.1), but this staining does not necessarily represent the active tissue depth.

17) How was it determined that the interval between the pink (ii) and yellow (iii) stars represents the soft tissue depth where pre-existing skeleton thickening occurs? This is a central concept and requires more detailed explanation.

We acknowledge that the concept of the thickening interval within soft tissues is a crucial part of this study and was not well explained in the original manuscript. This is explained in our previous study¹ and have now added more detailed explanation on lines 61-77.

18) The reconstructed TSS does not match the tissue thickness data from Vincent and Sheldrake (2025), how you reconcile this?

In the study of Vincent & Sheldrake (2025)¹, the measured tissue staining was 7.17 mm at the surface of the sample. Skeletal thickening was shown to occur to a depth of 4.04 mm within this tissue staining. We have now changed the starting position of the reconstructions to between the first observed skeleton and the fully formed skeleton in the CT-slices to better synchronise all four geochemical transects. The remeasured skeletal thickening interval is 3.8 mm, which matches the offset between the first porosity trough and Li/Mg peak (3.7 mm). We assume that the depth of skeletal thickening interval within soft tissues is relative to the total tissue depth and thus variations in the offset between porosity and geochemistry correspond tissue depth. This measured offset is less than the depth of tissue staining at the surface of the sample (see supplementary figure S.1), but this staining does not necessarily represent the active tissue depth. We have clarified this point on lines 213-215.

19) Please clarify that panel e refers to the calyx depth in *Siderastrea*. How confident are the authors that the measured structure corresponds to the calyx depth?

We have clarified this in the figure caption. We are confident as we can confirm the calyx depth visually using the reconstructed CT slices.

20) The assumption of a V-shaped calyx may not always hold: in some *Porites* species, the columella and adjacent pali can create a W-shaped profile, potentially increasing actual calyx depth relative to a simple V-shape.

We acknowledge that other corallite forms influence the interpretation of corallite depth. The method in this study is aimed towards massive scleractinian corals with a V-shaped calyx (*Porites*, *Siderastrea* etc). For corals with different corallite morphologies (i.e., W-shaped), further investigation is needed to interpret the corallite formation and thickening processes as done in our previous study¹. We have included this point on lines 358-361. We are currently working on a manuscript for *Diploria strigosa* sample which has a more complex, rugose corallite structure.

Reviewer 2

Vincent and Sheldrake examine how massive scleractinian corals allocate energy between skeletal growth and soft tissue formation, revealing that seasonal tissue thickness is influenced by environmental stress. The authors suggest that variations in soft tissue thickness directly drive skeletal extension and that offsets between geochemical cycles in the skeleton and growth bands may provide a means to reconstruct soft tissue thickness. The study finds that coral bleaching HotSpot values, which indicate climate change-related stress, are inversely correlated with growth band offsets, affecting future calcification. Additionally, the authors show that major environmental events, e.g. volcanic eruptions, may play a role in coral stress responses, offering insights into how corals have historically adapted to changing conditions.

This is an interesting paper on an important topic. The authors use a new method introducing a potential coral stress response indicator and test this novel indicator against recent warm/stress events. The findings are a very valuable contribution to the field and are of major interest to the community. Thus, the work is important and worthy of publication in a journal like *Communications Earth & Environment*. The conclusions are convincing, however there are some ways that the manuscript can be improved to strengthen the conclusions, and thus I would recommend that the authors revise the manuscript. Detailed comments are given below.

Detailed comments:

21) The authors reference their previously published paper from the same study site (line 60), which has focused on micro-CT analysis and the establishing of their method to analyze growth banding and classify microstructures and reconstructing skeletal porosity. This study has used the same core from 15 m water depth (referenced in Suppl. Fig.1 and reference number 53), plus a coral core from 5 m water depth at a second site. I am wondering why here only one of the cores is used for LA-ICP-MS analysis (15 m water depth core from NW-

Barbados). If laser data is not available for the shallower site, it would be helpful to add it or give a reason why the core from the deeper site has been chosen for this study. The authors suggest that their results could be very helpful for predicting coral response to anthropogenic stressors (such as high SST events). Thus, having data from shallower water, where water temperature is often showing higher variability (and more likely to be affected by temperature extremes as regularly captured by satellite SST observations) would be useful.

The second core of *Siderastrea* from our previous study (named W1 in Vincent & Sheldrake, 2025¹ was not used in this study because of the complicated growth history of the colony. The irregular extension rates between corallites, which we attributed to the shallow water-depth in which the colony grew (i.e., more variability in SST, insolation and wave energy), which meant that the CT slices reconstructed corallites at different stages of thickening, which dampened and masked the seasonal porosity signal. The signal was therefore not clear enough to constrain the peak and trough locations needed to calculate the offset which we interpret as tissue depth. This is an important limitation to this study, and we acknowledge that in the conclusions on lines 358-361 where we state that further advancements are needed with more complicated growth structures. Nevertheless, we believe the novelty of the method can be justified with sample HP1 that has a clearer growth history. We have added a statement into the manuscript stating why W1 was not used in the study on lines 134-138.

22) The Amazon and Orinoco Rivers significantly influence the study site due to its position at the southeastern margin of the Caribbean Sea. This is briefly noted in the Methods (line 463). Owing to this geographic setting, a pronounced seasonal influence on Barbados is expected as North Brazilian Current rings episodically transport waters originating from these rivers into the region. These riverine plumes are characterized by elevated sediment and nutrient loads, which may reach the study site during certain periods of the year. Is the coral Ba/Ca ratio potentially influenced by this transport of sediment-loaded waters, which also bring nutrients to the study site? Given the observed clear and recurring seasonal cyclicity in the coral Ba/Ca ratios, it is plausible that these ratios are modulated by the periodic arrival of sediment- and nutrient-rich waters from the Amazon and Orinoco Rivers. Alternatively, local processes that deliver Ba-enriched waters to the site could also contribute to the seasonal Ba/Ca signal. However, the timing and nature of the Ba/Ca variability are consistent with the established seasonal dispersal patterns of the Amazon and Orinoco River plumes in the eastern Caribbean. Therefore, I suggest to discuss the potential contribution of Amazon and Orinoco River discharge as a source of barium in greater detail within the main text, especially with respect to the timing of the seasonal cycle when comparing potential local vs. more distant sources of sediment/Ba input.

We acknowledge that these currents have the potential to influence Ba/Ca ratios and that this was not discussed in the original manuscript. We have now added more detail on the NBC rings on lines 408-423 in the description of the study location. Nevertheless, understanding the precise cause of the Ba/Ca signal is not the focus of our investigation and we state this on lines 423-424. For us, Ba/Ca acts as a stationary seasonal signal in the theca that is offset from the corallite microstructures, which is explained on lines 198-209.

minor comments:

23) line 337 - Figure 1: raw Ba/Ca and Li/Mg ratios (thin solid lines) look fine in the pdf but not bright enough when printed on paper.

We acknowledge that this figure was not clear enough and have changed this figure for better clarity. Please see Figure 3 in the new version of the manuscript.

24) line 90: 3- to 4 year cycles are reported in the offsets, which are linked to timing of rainfall and SST at the site. Earlier studies have shown potential links to El Nino Southern Oscillation in records from the Caribbean Sea in this frequency band.

We acknowledge that ENSO events could be related to the 3-to-4-year signals shown in our results. We do not specifically discuss this in the manuscript because we have no evidence/data to back this hypothesis but mention it in the study location description on lines 400-406.

Reviewer #3 (Remarks to the Author):

General comment

The study proposed a novel methodology for measuring past tissue layers on coral records. This approach combines microstructure and geochemical data over a 60 mm segment of a single coral core from the species *Siderastrea siderea*, collected from the coast of Barbados in the Caribbean Sea. The authors claim that this approach can be applied through a coral core to produce information about past tissue layers conditions and how it is influenced by environmental stressors on the coral. While the approach is intriguing and offers novel insights into paleoclimate and environmental reconstruction based on coral archives, it is essential that the necessary data is thoroughly tested before its implementation.

Major considerations

25) The primary concern with this approach is its lack of robustness, due to its basis on a single geochemical and microstructure record from a single colony. I am aware that geochemical measurements can result in significant costs, however, for a study proposing a novel approach that could be of considerable benefit to the scientific community, I believe it would be advisable for the authors to collect additional data. As the authors are working with small sections of coral cores, replication is very feasible; at least 5 to 10 replicas could be obtained. Even though new field work for the collection of additional coral cores can be complex, it is still possible to replicate the analyses using the same coral slab. For example, they could provide five geochemical transects along five parallel theca walls concomitants with porosity measurement on the same coral slab. Additionally, reference number 53 (Vincent, J. & Sheldrake 2025), which is from the same authors, presents micro CT-Scan data on two *S. siderea* corals. Why didn't they explore at least these two records? The authors could even get 3 to 5 replicas on geochemical and porosity data from each coral core to increase the robustness of their interpretation. From my own experience of replicating geochemical measurement on different theca of the same coral slab, I know that slight variations in results can be expected for a number of reasons. This is a key consideration in this new approach. The primary question to be addressed by this study is to determine whether different paired geochemical-porosity records yield the same relationship.

If so, we can be confident that this new approach can be used to reconstruct previous tissue layers and infer possible stressors from different records of *S. siderea*.

We acknowledge the lack of replicates within the previous version of the manuscript. In the new version of the manuscript, we have analysed three additional LA-ICP-MS transects on HP1 core so that we have in total analysed four different thecae to increase replicability. All three additional transects have double the spatial-temporal resolution, with analyses spots spaced at 180-microns. We thank the reviewer for this comment, as it can be seen in Supplementary Figure S.6 and S.7 that the individual transects are different, and we have shown that this issue is worse if we down-sample the data. However, once we combine 3 transects together, we see the average value converges irrespective of what combination is used, and there is no difference if we combine all four transects together. Hence, we are confident that the replication provides us with the true sample-level geochemical signal. Although the four individual transects are slightly offset from each other, they each show the same cyclical pattern that we are confident the average is a true signal. We show that this signal is stable, even if the data is sampled at a lower resolution and what is most important is to sample across multiple thecae. We state this point on lines 244-247.

The second core from our previous study was not used in this investigation due to the irregular extension rates between corallites which dampened the seasonal porosity signal. We attributed the poor growth banding to relatively less stable conditions at shallower depths (i.e., more variable SST, wave energy and insolation). We realise that this is not justified in the original manuscript and has been added into the intro on lines 134-138.

Without clear growth bands, the reconstruction of tissue depth is more complicated as requires further method development. Nevertheless, we believe that this does not invalidate the utility of the method for samples with clear growth banding. Future development of the method could allow the concept to be applied to samples with more complex growth histories, and we mention this in the conclusions on lines 358-361. However, to introduce the method and based on demands of clarity from reviewers, we believe it is better to focus on one clearer sample.

26) A second and also major issue is the quality of the geochemical data produced by LA-ICP-MS, which is known to be noisy. The authors claimed that the high RSD wouldn't impose a limitation on the interpretation of geochemical cycle. I have some concerns about that, which is aggravated by the lack of replications. A noisy record could drift the smooth line when assuming the highest and lowest values of the record. That can compromise the whole history raised by the authors. A further question to be addressed is that of the rationale behind the authors' decision to not utilize Sr/Ca ratios as a proxy for SST, given the long- and well-established nature of this method. The selection of Li/Ma and Ba/Ca ratios was predicated on which criteria?

We acknowledge that the y-axis error (i.e., RSD of the element ratios) can influence the x-axis error (i.e., by drifting the smooth line) and we appreciate this concern, especially with the lack of replications as mentioned in comment 25.

The 2RSD values we report in the original version of the manuscript align with the limited number of publications that uses JCp-1-NP reference material to gauge precision and accuracy of LA-ICP-MS analyses²⁻⁸. We have added these references into the new version of the manuscript on lines 532-533. The updated 2RSD values of JCp-1-NP for Li/Ca, Mg/Ca and Ba/Ca are 11.2 %, 8.3 %, and 8.5 %, respectively (n=36), which use the mean. We are currently collaborating on a paper which characterises the precision of the JCp-1-NP pellet⁸, using different LA-ICP-MS parameters across different laboratories between 2022 to present. Our results from this collaboration show a mean 2RSD of 15.78 (n = 226), 11.43 (n

= 250) and 13.9 % (n = 225) for Li/Ca, Mg/Ca and Ba/Ca, respectively. These results show that the quality of our results is replicable and reliable. The slight variability in reproducibility of JCp-1-NP likely underlines the heterogeneity of the natural nano-powdered Porites sample.

To quantify the influence of y-axis error on the x-axis, we performed bootstrapping (n = 50,000) on the Li/Mg and Ba/Ca ratios from all transects using the analytical session specific 2RSD of JCp-1-NP. Each spot was resampled according to the 2RSD, and the LOESS smoothing was applied to fit 50,000 smoothed regressions. The final results are shown in Figure 2 in the new version of the manuscript and show neglectable influence on the x-axis error despite significant differences in the amplitudes of the cycles. The dashed lines in Figure 2 represent the 5th, 50th and 95th percentiles. Given our study is focused on the position of the peaks, and not the magnitudes, we are confident that the 2RSD will not influence our conclusions.

Both Li/Mg and Ba/Ca ratios were chosen as they show cyclicity with similar wavelengths as the porosity, which varies with seasonality. Li/Mg is well-established SST proxy which shows a negative correlation to SST. Sr/Ca was measured in both LA-ICP-MS sessions but was not chosen in this study due to the poor clarity in the seasonal signal.

27) A personal recommendation to the authors is to consider adopting a more conventional approach, such as micromilling techniques for producing powder that can easily dissolved in HNO₃ and analyzed by ICP-MS, which will probably result in much more precise geochemical data. The coral HP1 exhibits a growth rate that permits the generation of more than 12 samples per year through micromilling, by using a sampling resolution of 0.5 mm. This resolution is sufficient to report the full geochemical seasonality of the coral (see the work of Maupin et al., 2008, <http://dx.doi.org/10.1029/2008GC002106>). I believe that would increase the robustness of the geochemical data of this study. The author should also mention the growth rate of the colony, they have this information.

We thank the reviewer for their recommendation and recognise the value of solution-based ICP-MS, especially when accurate and precise data are required to reconstruct environmental variables from geochemical ratios. The LA-ICP-MS method used in this study was developed for higher spatiotemporal resolution than possible with micromilling, which is required to characterise the shape of each peak and trough, and thus quantify the FWHM and seasonal extension. The mean extension rate of our sample was 5.89 mm/year, now written on lines 194-195 in the manuscript. Our method in the original version used 17 analyses points per cycle/year to fit a Gaussian curve. The recommended method (comment 27) would reduce the resolution to 12. With the additional transects measured to increase replicability (see response to comment 25), we have increased the spatial-temporal resolution to 34 points per cycle. It is better to have more points per curve as shown by our down sampling results (see response to first major comment (25)).

In addition, the method used in our study ablates a surface area of 60-microns of CaCO₃ which equates to approximately 4 days of coral growth. Micromilling 0.5 mm of CaCO₃ however averages 30.2 days, averaging out a larger time period which reduces the spatial resolution and also averages more time, obscuring shorter-term stress events.

The present study, which proposes a novel methodology for the assessment of past tissue thickness, lacks a thorough examination of the aforementioned issues. Consequently, it is not sufficiently robust to be published in Nature Communications Earth & Environment.

Additional Important points:

27. Please, provide a figure with the coral core slab image and the location where the geochemical and porosity data were acquired. This is very important for what you are proposing.

We have added a supplementary figure of the LA-ICP-MS transects (see Supplementary Material S.5).

28. The AIMS the study “how energy is allocated during seasonal growth” and “how stress influences energy allocation”, it is not appropriate for the study. The authors should first establish the approach and not dive into more complex questions like those one proposed.

We acknowledge that we are overstepping the level of interpretation that we could make and so based on this comment and those of the other reviewers, we have reframed the paper to remove the focus on “energy”. We now focus the paper of reconstructing tissue depth and coral stress/health, and we hope it will be further adapted in the future for different coral species and complex growth histories.

29. The authors should address properly why they observed 10 geochemical cycles against 9 porosity cycles.

This was explained in the referenced study Vincent & Sheldrake (2025)¹. We have rewritten this section for better clarity on lines 177-227.

30. Li/Mg and Ba/Ca are mostly in phase through the record. First, I would compare this with the local sea surface temperature (SST) and precipitation to check if the pattern is also happening in real environmental conditions. I would also make a crossplot of the Li/Mg and Ba/Ca geochemical data, as well as a crossplot of the SST and precipitation data to check their covariance. I would also suggest that the authors include a figure showing the climatology of SST and precipitation for the location, since they are assuming that the porosity index, Ba/Ca and Li/Mg are related to these variables. This would help them understand the index better. There are available data on precipitation and SST for the region and a time series from 2022 to 2012 would make the work much better.

Although it appears to be in phase in Figure 1 of the original manuscript, the Li/Mg and Ba/Ca are offset/lagged which can be seen in Figure 2a of the original manuscript. We have now changed this Figure (see Fig. 3 in the new version) which shows the offset between Li/Mg and Ba/Ca more clearly by the differences in the size of the pink arrows. Additionally, we have added supplementary Figures (S.2, S.3, and S.4) which includes the SST and precipitation data. These figures show that there is no correlation between Li/Mg and Ba/Ca due the offset between ratios, and a moderate correlation between SST and precipitation ($r = 0.69$). The peaks and troughs in the SST and precipitation data show variable offsets.

The aim of this paper is not to examine the validity of Li/Mg and Ba/Ca as SST and nutrient/precipitation tracers. Instead, we use them to track seasonality which is known to control the formation of growth bands. For example, we are aware that at tropical SSTs, there are large variations in the calibration of Li/Mg with SST, making it challenging to reconstruct absolute values¹³. Nevertheless, each calibration shows a negative correlation with SST which gives us confidence to believe that the peaks and troughs correspond to annual low and high SSTs respectively.

31. Why did the study suddenly change to focus on the volcanic event in Barbados? You are still trying to work out the approach! Moreover, there's a lot of speculation about this topic, which could only be addressed by investigating longer coral records that registered the volcanic event in 1979.

The sample was originally collected to investigate volcano/coral interactions after the eruption of La Soufrière, St. Vincent in April 2021. The relationships discovered in this manuscript were made as part of understanding the sample in the context of volcanic ash-coral interactions. The volcanic eruption was a large event which is shown to influence coral physiology¹⁴. We show that the eruption had implications on the relationships shown in this study which we believe to be important for coral stress/health. We use the eruption to highlight that tissue depth can be affected by a variety of climatic and environmental events.

We realised that this was not clear in the original manuscript, so we have added context in the introduction on lines 128-130 and 273-280. It is clear from our results however that the eruption influenced calcification rates and tissue depth and its inclusion in the manuscript is justified.

Specific comments:

32. Lines 27-31. As the manuscript focus in coral skeletal structure I think some very important citations are missing here. It would be nice to cite the first discoveries on the topic, for example the works of:

Knutson et al 1972 [<http://dx.doi.org/10.1126/science.177.4045.270>]

Buddemeier et al 1974 [[http://dx.doi.org/10.1016/0022-0981\(74\)90024-0](http://dx.doi.org/10.1016/0022-0981(74)90024-0)]

We have added these citations into the new version of the manuscript on lines **34-35**.

33. Line 57. Define FWHM at your first entrance in the text, readers might not be familiar with the term.

This term was explained in the methods section of the original manuscript. We have now added the definition to the introduction on lines 149-151 and methods section on lines 565-586.

34. Lines 337-347. The caption of figure 1 is hard to follow, I suggest reformulating the figure for a better visualization of what the authors want to say. The blue and red dash lines are confusion. The authors presented the porosity record twice and I don't see a practical reason. It must have a better way to show the data. Moreover, they mentioned a vertical "grey dashed line" which is not present in the figure.

We acknowledge that Figure 1 was not clear. This figure has been remade for better clarity. Please see Figure 3 in the new version of the manuscript.

35. Line 74: The authors should specify which extreme they are mentioning. Are they mentioning the first highest value? If so, I suggest to insert a text in the Fig.1 A and B informing that this first value is interpreted as July 2022, that is why I am supposing that the core was collected in 2022, there is no information of the year of collection in this manuscript.

This information can be found on line 73 of the original manuscript and has been on lines 181-186. We have also rephrased the term extreme to peak and trough for simplicity.

36. 76: Again, the term extreme without specification. I suggest the author replace this term by "lowest or highest peak".

Please refer to response 35.

36. 77: should be "Fig. 1a"

The figure references in the text have been updated.

37. 78: there is no "Fig. 1c"

Please refer to response 36.

36. 77: should be "Fig. 1a"

Please refer to response 36.

38. Lines 80-84: That is truly hard to sell. The text is confusing, for example, the scale they use in the figure did not help the readers visualize where the Li/Mg peak is, which is even more complicated because there are one peak from the smooth and another more positive value from the raw data. The previous issue is simple to fix. The main problem is that they observed that the distance between the most recent porosity trough and Li/Mg peak matched the microstructure of what they call "thickening". However, this cannot be taken as absolute truth – it needs to be tested again.

The text has been rewritten for clarity on lines 66-77 and 177-227 of the new version of the manuscript for better clarity.

39. I suggest that the authors adds the supplementary figure 1 to the main text figure 1 and adds it as a zoom detail at the top of the record. That would make it much easier to understand their reasoning. It's tiring going back and forth to check the figure while reading the text.

We acknowledge that supplementary figure 1 in the original manuscript is a key figure and thank the reviewer for highlighting its need in the main text. We have now added this to the main text as Figure 1.

40. Authors should improve the provide and age model for figure 1 at the top of the figure. That will help to clean up the excessive labels at the top of the figure.

We have remade this figure (see Figure 3 in the new version of the manuscript).

41. Lines 152-186. That is a very interesting topic, but highly speculative. The authors should drop this topic and not include it in the current manuscript. It should be left for another assessment where replications are properly done and longer records are available, which include another volcanic event.

Please refer to responses to comments 25 and 31.

1. Vincent J, Sheldrake T. Micro-CT analysis reveals porosity driven growth banding in Caribbean coral *Siderastrea siderea*. *Sci Rep*. 2025;15(1):6063. doi:10.1038/s41598-025-90125-w
2. Nambiar R, Kniest JF, Schmidt A, Raddatz J, Müller W, Evans D. Accurate measurement of K/Ca in low-[K] carbonate samples using laser-ablation sector-field inductively coupled plasma mass spectrometry. *Rapid Communications in Mass Spectrometry*. 2024;38(5). doi:10.1002/rcm.9692
3. McCormick CA, Corlett H, Clog M, et al. Basin scale evolution of zebra textures in fault-controlled, hydrothermal dolomite bodies: Insights from the Western Canadian Sedimentary Basin. *Basin Research*. 2023;35(5):2010-2039. doi:10.1111/bre.12789
4. Reuer MK, Boyle EA, Cole JE. A mid-twentieth century reduction in tropical upwelling inferred from coralline trace element proxies. *Earth Planet Sci Lett*. 2003;210(3-4):437-452. doi:10.1016/S0012-821X(03)00162-6
5. Jochum KP, Garbe-Schönberg D, Veter M, et al. Nano-Powdered Calcium Carbonate Reference Materials: Significant Progress for Microanalysis? *Geostand Geoanal Res*. 2019;43(4):595-609. doi:10.1111/ggr.12292
6. Förster F, Flöter S, Sauzéat L, et al. Volcanic ash leaching alters the trace metal distribution within the coral holobiont of *Stylophora pistillata*. *EGUsphere[preprint]*. Published online May 5, 2025. doi:10.5194/egusphere-2025-1713
7. de Winter NJ, Killam D, Fröhlich L, et al. Ultradian rhythms in shell composition of photosymbiotic and non-photosymbiotic mollusks. *Biogeosciences*. 2023;20(14):3027-3052. doi:10.5194/bg-20-3027-2023
8. Flöter S, Förster F, Vincent J, et al. A long-term study of the reference material JCp-1-NP: new and compiled LA-ICP-MS elemental compositional data. *Scientific Data[in review]*. Published online July 2025.
9. Carricart-Ganivet JP, Vásquez-Bedoya LF, Cabanillas-Terán N, Blanchon P. Gender-related differences in the apparent timing of skeletal density bands in the reef-building coral *Siderastrea siderea*. *Coral Reefs*. 2013;32:769-777. doi:10.1007/s00338-013-1028-y
10. Barnes DJ, Lough JM. Systematic variations in the depth of skeleton occupied by coral tissue in massive colonies of *Porites* from the Great barrier reef. *J Exp Mar Biol Ecol*. 1992;159(1):113-128. doi:10.1016/0022-0981(92)90261-8
11. Fowell SE, Sandford K, Stewart JA, Castillo KD, Ries JB, Foster GL. Intrareef variations in Li/Mg and Sr/Ca sea surface temperature proxies in the Caribbean reef-building coral *Siderastrea siderea*. *Paleoceanography*. 2016;31(10):1315-1329. doi:10.1002/2016PA002968
12. Förster F, Reynaud S, Sauzéat L, Ferrier-Pagès C, Samankassou E, Sheldrake TE. Increased coral biomineralization due to enhanced symbiotic activity upon volcanic ash exposure. *Science of The Total Environment*. 2024;912:168694. doi:10.1016/j.scitotenv.2023.168694

Reviewers' comments:

Reviewer #1 (Remarks to the Author):

General comments

We thank reviewer 1 for their critical but constructive comments which have helped improve the clarity and message of our manuscript.

1) Vincent and Sheldrake have done a very in-depth review of the manuscript, and I commend their thorough attempt to address all reviewer comments. However, I feel that some of the implemented changes have not improved the manuscript in the best possible way, particularly in the Introduction.

The Introduction remains too long and unfocused. It mixes background information, methods, results, and discussion, and thus reads more like a mini-review combined with a results preview. The expected logical structure: background, knowledge gap, and hypothesis/aim is not clearly evident. Instead, the text oscillates between general background, detailed methodology, and even findings. Several paragraphs seem to belong more appropriately in the Methods or Results sections.

We acknowledge that the introduction was poorly structured. To address this, we have shortened and restructured the introduction to make it more concise. The first two paragraphs concern the background to the manuscript (lines 22-72); the following two paragraphs (lines 73-102) present the knowledge gap in respect to understand and reconstructing tissue depth; and the final two paragraphs (lines 103-134) present the aims of the manuscript.

2) I am also concerned about the offset data, which no longer appears to track the hotspot data as clearly as before. This change weakens the central narrative, as it is less obvious that the data support the proposed mechanism. A scatter plot or correlation analysis might help clarify this relationship.

We have added a scatter plot to the new version of the manuscript (Fig. 5) to better show the relationship between tissue depth and thermal stress (i.e., offset and HotSpot values). We acknowledge that there is some scatter in this relationship which we attribute to the complexity of stress in the marine environment (i.e., global vs local stressors), as evidenced by the impact of the volcanic eruption and explained in the caption of Figure 5. Nevertheless, it is clear from the scatter plot that thermal stress is driving the underlying trend in the offset, and this gives us confidence that the offset is reconstructing tissue depth.

The difference between version 1 and version 2 of the manuscript, is related to the fact that individual corallites respond to stress differently which manifests in differences in tissue depth and extension rates. This would explain why tissue depth is not homogenous across the coral surface and why the growth surface is not perfectly

planar. By averaging the four theca profiles we present an average offset value for the sample.

Specific comments

3) Line 8 – Please avoid the use of “/”.

We have removed the “/” on line 8.

4) Line 16 – Suggest replacing future with subsequent.

We have replaced the “future” with “subsequent” on line 17.

5) Line 36 – Generally true, though some exceptions have been reported

(see: <https://www.ingentaconnect.com/content/umrsmas/bullmar/2000/00000066/0000001/art00020>

).

We have added that banding generally reflects variations in skeletal extension and thickening rates on lines 34. Thank you for this suggestion, we have added the reference to the manuscript.

6) Line 50 – The main issue here is whether tissue thickness is variable.

Tissue thickness, as stated and referenced on lines 80-81 of the previous version of the manuscript, is variable and adds to the complexity of bio-smoothing and interpreting coral geochemical proxies.

7) Line 52 – This line could be removed.

We have removed this line from the introduction and inserted it into the conclusions on lines 327-329 as an outlook summarising that it may be an important issue to consider.

8) Line 61 – Paragraph feels too detailed; reads more like methodology/results.

We have synthesised this paragraph on lines 103-111 and written it in context to the hypothesis/aims of manuscript. In addition with the information in Figure 1, we believe this clearly and systematically explains the key concept of the offset and its link with tissue depth.

9) Line 83 – Suggest removing “negative”.

We have changed this sentence on line 46 so it no longer includes the term “negative”.

10) Line 87 – The relative impact of volcanic eruptions is likely minor among reef stressors.

We included volcanic eruptions as a local stressor as it is relevant to our sample in this study. We agree that the relative impact of volcanic eruptions is minor compared to factors such as heat stress, which we believe our results show in the new scatter plot in Figure 5.

11) Line 110 – TD not defined.

We have defined tissue depth (TD) on line 73 in the new version of the manuscript.

12) Line 128 – This section feels more appropriate for Methods or Supplementary Information.

We have added most of this information to the methods on lines 362-363 and 416-424 but have adapted the first two sentences on lines 120 to give context to the following paragraph and the rest of the manuscript.

13) Line 141 – Could be condensed.

We have condensed this paragraph, now on lines 120-172.

14) Line 166 – Suggest adding “coral”.

Thank you for this suggestion, we have added “coral” into the subtitle on line 136 of the new manuscript.

15) Line 186 – The note “(note the Li/Mg is reversed for better comparability)” is redundant; already stated in the figure caption.

We have removed this sentence from the new manuscript.

16) Line 200 – Most of this information is only needed in the figure caption.

We have removed repetitive information from the main text of the manuscript.

17) Line 206 – Is the offset consistent when converted to the time domain? Does the offset in SST and rainfall align with that in Li/Mg and Ba/Ca?

We have been very careful throughout the manuscript to refrain from presenting and assessing environmental proxies using coral geochemistry. Based on a mean average extension rate of 5.9 mm/yr, the difference (in microns) between the Li/Mg and Ba/Ca offset represents approximately 46 days. We are unaware of either Li/Mg or Ba/Ca being interpreted at such high resolution, and so are careful to interpret the geochemical records at such a high temporal resolution. We have added a supplementary figure (S.4) and conditioned the sentence in the manuscript on lines 184. Furthermore, we have changed a sentence in the introduction on lines 123-125 to state that we use Ba/Ca to “represent” the rainy season and not “reconstruct” it. Given the complexities of surface hydrology and nutrient availability, reconstructing rainfall beyond a seasonal signal from Ba/Ca is unfeasible.

16) Line 214 – What could this represent, then?

We have now made it clear in the manuscript on lines 216-219 that the staining on the surface likely represents the tissue layer that is actively calcifying, as well as deeper organic material, such as tissue undergoing necrosis or other endolithic species.

19) Line 232 – Please avoid using “This” alone as a subject (“This variation...” should be specified).

We have carefully gone through the manuscript and updated sentences where “This” appears alone as a subject.

20) Line 256 – The pattern is not evident for all years; 2013 and 2016 yes, but not others. Consider quantifying the relationship (e.g., scatter plot between thermal stress and offset).

We have added a new scatter plot between the offset and HotSpot values (i.e., thermal stress and tissue depth) to better show their general relationship. For further discussion, please see our response to the general comments at the start of this review. We have changed this sentence to reflect more clearly what is shown in Figure 4a and in the new scatter plot Figure 5 on lines 281-284, 321-325, and 339-340.

21) Line 279 – In the original version, the offset decreased after 2020, whereas it now increases, yet the explanation remains unchanged. This inconsistency undermines the argument — how can the same explanation apply if the trend is reversed?

In both the original manuscript and in the revised version the offset increased after 2020 (i.e., the cold/dry-season of 2021), which we relate to the volcanic eruption. Therefore, the explanation has remained unchanged. We appreciate that the decrease in 2020 is not so pronounced in the revised version compared to the original version, but please see our response to the general comments at the start of this review concerning the scatter plot.

22) Line 321 – Please rewrite for clarity; currently hard to follow.

This line has been rewritten on lines 361-362 in the new version of the manuscript.

Reviewer #2 (Remarks to the Author):

23) I have no further comments and accept the manuscript in its revised form after carefully evaluating the authors response.

We thank the reviewer for their comments in the first round of reviews which contributed significantly to the manuscript.

Reviewer #3 (Remarks to the Author):

We thank the reviewer for re-reading our manuscript, which meant that we have made the narrative even clearer and expanded the context of the discussion to include issues with mis-interpreting relationships between seasonal variations in coral porosity/density and geochemistry.

24) The authors have improved the manuscript by doing three additional LA-ICP-MS transects on HP1 core. This represents an improvement; however, I still find the interpretation to be somewhat speculative. In my view, proposing a new proxy based on a single record, especially when combining multiple analytical approaches, warrants greater caution.

As suggested by reviewer 3 in the first round of reviews, we have increased the robustness of our study by analysing three additional LA-ICP-MS transects on three

independent thecae records. As stated in comment 25 from the first round of reviews, each theca transect should be treated as one independent record. The goal of our study is to calculate the offset that manifests at the surface of the sample, and to explore what influences it. We have been able to establish a relationship between the offset distance and physiological stress, which is supporting by the new scatter plot (Fig.5). Additionally, we want to make it clear that we propose a method to quantify/estimate coral tissue depth or more generally coral health, and to explore how this relates to environmental changes. At no point has the manuscript proposed been presented as the development as a proxy to reconstruct environmental variables.

The HP1 core provides an exceptional archive of linear growth, making it uniquely suited for this study. Currently, the method proposed in our manuscript is limited to samples with simple growth histories as already mentioned on lines 360-363 of the previous version of the manuscript. It is not uncommon for coral-based studies to use one coral colony to interpret variations in environmental conditions¹⁻⁴. We therefore do not believe that using multiple replicates from the same colony limits the value of our results.

25) I acknowledge the authors' efforts to make the study as robust as possible by addressing all the comments the reviewers have made. Nevertheless, there appears to be a significant conceptual error regarding the Li/Mg proxy that could compromise their interpretation. Specifically, they state in lines 183–186:

"Given the sample was collected in July 2022, transitioning from C/D-season to W/W-season, we interpret the closest porosity trough to the growth surface (orange star – Fig.3) to represent the C/D-season with low SST's when investment of energy into linear skeletal extension is expected to be lower."

Reviewer 3 has misinterpreted Figure 3 which has led to a misunderstanding of the manuscript's narrative (comments 25-30). We consider this an important insight and so have added a sentence discussing this on lines 159-161 and in the conclusion on lines 366-368.

In Figure 3, we show corallite porosity, Li/Mg and Ba/Ca ratios on a spatially synchronised x-axis (mm). Reviewer 3 has interpreted the x-axis of this graph as time which does not account for 1) the spatial difference between the theca (i.e., Ba/Ca and Li/Mg ratios) and the base of the corallite (cup-like structure), and 2) the skeletal thickening interval beneath the corallite (i.e., location of growth band formation) which was introduced and cited on lines 63-81 in version 2 of the manuscript, as well as Fig. 1. Both these spatial offsets need to be accounted for when interpreting spatially combined growth and geochemical records. To reinforce this message, in the first sentence of the caption in Fig. 3 we have added the term "theca" twice preceding the geochemical ratios. The orange arrows in Figure 3 are there to guide the reader on how to interpret the records, showing a deeper/negative shift of the geochemical profiles on the x-axis which account for these spatial offsets. The arrows also show the systematically changing offset between corallite porosity cycles and geochemical

profiles which we relate to variation in tissue depth. The dashed vertical lines extend from the growth bands into the Li/Mg and Ba/Ca records to more clearly illustrate the cyclic offset between porosity and geochemical records. The Li/Mg ratios are inversed in Figure 3 for clarity, which were stated in both the text on line 86 and in the figure caption.

Correctly interpreting this figure will alleviate concerns outlined in the subsequent comments (25-30). Li/Mg and Ba/Ca are inversely correlated supporting increasing SSTs and rainfall in the warm/wet-season as shown in Supplementary Figures S.2, S.3, S.4 which were requested by reviewer 3 in comment 30 from the first round of reviews. Li/Mg is decreasing (i.e., increasing SST), and Ba/Ca is increasing (i.e., increasing rainfall) toward the growth surface which agrees with the context of the sample which was sampled during the transition between cold/dry-season to warm/wet- season.

26) However, the peak corresponding to lower Li/Mg values (orange star) indicates higher SST. How, then, do the authors interpret this as representing a cold/dry season? Low Li/Mg ratios thermodynamically correspond to higher temperatures, as demonstrated in several studies, including Cuny-Guirriec et al. (2019), which the authors cite, rather than the opposite.

The orange star indicates the position of the porosity trough, which is extended vertically by the dashed lines into the geochemical profiles. The orange arrows indicates the direction of spatial offset of the Ba/Ca and Li/Mg troughs needed for the porosity and geochemical records to be temporally synchronised. Li/Mg is inversed in this figure.

27) Again, in lines 189 to 193

“On this basis, high porosity bands correspond to the W/W-seasons characterised by fast skeletal extension rates (ranging between 3.56 to 5.17 mm/season), whilst low porosity bands correspond to C/D-seasons with slower extension rates varying between 1.25 and 2.17 192 mm/season (Fig.4).”

See response to comment 25 and 26.

28) That is not what the Li/Mg data indicates. They show higher Li/Mg ratios, which correspond to colder SST.

See response to comment 25 and 26.

29) And again, in lines 200-204

“The position on the porosity troughs (grey stars – Fig. 3) are projected into the Li/Mg and Ba/Ca profiles (grey dashed lines – Fig. 3) to highlight these offsets between the porosity troughs and corresponding Li/Mg peaks and Ba/Ca troughs which are hereafter termed C/D-season offsets (note that y-axis is reversed for Li/Mg in Fig. 3). The porosity

peaks show a similar negative offset with Li/Mg and Ba/Ca cycles which is hereafter termed W/W-season offset.”

See response to comment 25 and 26.

30) The grey lines are not indicating Ba/Ca thoughts!

31) This is not a minor oversight. The misinterpretation affects all subsequent sections that rely on the Cold/Dryer and Warm/Wetter assessments, which at some point compromising the overall narrative of the TD proxy application.

See response to comment 25 and 26.

References

1. Chalk, T. B. *et al.* Mapping coral calcification strategies from in situ boron isotope and trace element measurements of the tropical coral *Siderastrea siderea*. *Scientific Reports 2021 11:1* **11**, 1–13 (2021).
2. Jiang, Q. *et al.* Coral Ba/Ca and Mn/Ca ratios as proxies of precipitation and terrestrial input at the eastern offshore area of Hainan Island. *Journal of Ocean University of China* **16**, 1072–1080 (2017).
3. Li, X., Zhang, L., Liu, Y. & Sun, W. The Ba/Ca record of coral from Weizhou Island: Contributions from oil-drilling muds and the winter monsoon. *Mar Pollut Bull* **174**, 113317 (2022).
4. Brenner, L. D., Linsley, B. K. & Dunbar, R. B. Examining the utility of coral Ba/Ca as a proxy for river discharge and hydroclimate variability at Coiba Island, Gulf of Chirquí, Panamá. *Mar Pollut Bull* **118**, 48–56 (2017).

Reviewer #4 (Remarks to the Author):

General comments:

The manuscript entitled "Reconstructing environmental stress in corals by quantifying tissue depth using skeletal microstructural offsets" explores the potential of using geochemical proxies in the *Siderastrea siderea* coral skeleton to reconstruct past tissue-layer depth and assess environmental stress on corals. This study presents an interesting and potentially valuable approach for the biogeochemical community and for researchers utilising coral geochemical proxies. However, the manuscript still contains uncertainties regarding the ability of the proposed tool/approach to effectively reconstruct past tissue-layer depth. Therefore, I recommend minor revision of the manuscript, as outlined in the detailed comments below. In addition, Figure 3 requires improvement before the manuscript can be accepted for publication to prevent possible misunderstandings among readers.

Note:

My review focuses on two major concerns, as requested by the editor:

1. A possible conceptual error and misinterpretation of the coral skeletal proxy Li/Mg
2. The adequacy of the number of records to support the proposed tool for reconstructing past tissue depth and assessing environmental stress on corals

Concern (1) Possible conceptual error and misinterpretation of the coral skeletal proxy Li/Mg – Figure 3:

In my review, I did not identify any conceptual errors or misinterpretations of the coral Li/Mg proxy. As described in the manuscript (L41-42, 119-120), coral Li/Mg can serve as a temperature proxy. It is well established that coral Li/Mg inversely correlates with temperature (i.e., higher Li/Mg values correspond to lower temperatures) (e.g., Hathorne et al., 2013; Fowell et al., 2016). However, Figure 3 may lead to potential confusion among readers. The authors note that the Li/Mg record is plotted in inverse orientation on the y-axis, and that the offsets are indicated by symbols (stars), grey dashed lines, and arrows in both the figure and its caption. Although L157-164 provide detailed explanations, the figure may still be misleading to some readers.

To mitigate this risk and better highlight the interesting findings, I recommend the following improvement to Figure 3: In the current figure, the orange and grey stars indicate cold/dry seasons, and the offsets between porosity and chemical profiles are illustrated by grey dashed vertical lines and arrows. While these annotations help identify offset positions, they might cause confusion because the stars (denoting cold/dry seasons) appear only in the corallite porosity panel. I suggest that the authors also add similar stars in the Li/Mg (and Ba/Ca) panels to indicate cold/dry seasons for each cycle. This modification would more clearly demonstrate the offset relationships and the correspondence of seasonal cycles among corallite porosity, Li/Mg, and Ba/Ca records.

We thank the reviewer for their suggestions concerning Figure 3 of the manuscript and added open stars on the Li/Mg and Ba/Ca plots to guide the eye of the reader.

Concern (2) Number of records supporting the proposed "tool": In the abstract (L18–19), the authors state that this study "provides a tool to reconstruct past tissue depth and reconstruct environmental and ecological stress." The approach presented is indeed novel and has strong potential for reconstructing past tissue depth using *Siderastrea siderea* corals. However, since the study is based on a single coral core (n = 1), it remains essentially a case study and does not yet justify being described as a new "tool." Although the revised manuscript includes additional geochemical data from three transects, all transects were derived from the same coral core. These can be regarded only as within-core replicates rather than independent confirmations. Furthermore, it remains uncertain whether the specimen (HP1) used in this study is representative of *S. siderea* colonies with V-shaped corallites in the study area. To validate the proposed approach, a multi-colony-based study would be required to statistically confirm its reproducibility and robustness. Therefore, I recommend that the authors replace the term "tool" with "approach" or "concept" throughout the manuscript (not only in the abstract but consistently across the entire text). In addition, it would strengthen the manuscript to explicitly acknowledge the need for and encourage a future multi-colony validation study to test the robustness of this approach.

We acknowledge that our study presents a case-study and that further investigation is required at both an inter-colony and intra-colony scale to be applied as a universal tool to reconstruct tissue depth and thus environmental and ecological stressors. This has been emphasised in the abstract on lines 11 and 12, 18-21 and in the conclusion on lines 357-370. In addition, we have removed all suggestion as the proposed approach as a definitive tool throughout the manuscript.